# Illuminating the photon content
# of the proton within a global PDF analysis

**Valerio Bertone[1], Stefano Carrazza[2], Nathan P. Hartland[1] and Juan Rojo[1]**

**1** Department of Physics and Astronomy, VU University, NL-1081 HV Amsterdam,
and Nikhef Theory Group, Science Park 105, 1098 XG Amsterdam, The Netherlands
**2** Theoretical Physics Department, CERN, CH-1211 Geneva, Switzerland

## Abstract

**Precision phenomenology at the LHC requires accounting for both higher-order QCD and electroweak corrections as well as for photon-initiated subprocesses. Building upon the recent NNPDF3.1 fit, in this work the photon content of the proton is determined within a global analysis supplemented by the LUXqed constraint relating the photon PDF to lepton-proton scattering structure functions: NNPDF3.1luxQED. The uncertainties on the resulting photon PDF are at the level of a few percent, with photons carrying up to $\simeq 0.5\%$ of the proton's momentum. We study the phenomenological implications of NNPDF3.1luxQED at the LHC for Drell-Yan, vector boson pair, top quark pair, and Higgs plus vector boson production. We find that photon-initiated contributions can be significant for many processes, leading to corrections of up to 20%. Our results represent a state-of-the-art determination of the partonic structure of the proton including its photon component.**


# 1 Introduction

Recent progress in the computation of higher-order QCD corrections to LHC processes is such that the current state-of-the-art accuracy is NNLO, with even $N^3LO$ calculations available in some relevant cases (see Ref. [1] for a review). At this level of theoretical precision, the inclusion of electroweak (EW) corrections becomes phenomenologically relevant. With this motivation, NLO EW corrections to hard-scattering matrix elements have been computed for many LHC processes, including single and double vector boson, inclusive jets and dijets, and top quark pair production, among others [2–11]. Alongside progress made in process-specific calculations, the automation of NLO EW calculations [12–14] has also advanced significantly.

In order to make the most of these developments in the calculation of higher-order QCD and EW corrections, equivalent progress in the determination of the parton distribution functions (PDFs) of the proton [15] is vital. In this respect, most of the recent global PDF analyses [16–20] are indeed based on NNLO QCD theory. On the other hand, PDF analyses that include QED and weak effects and a determination of the photon PDF are scarcer [21–24]. Such QED PDF sets are required by consistency once EW corrections to matrix elements are included, as well as to account for the effects of photon-initiated (PI) subprocesses.

Indeed, the inclusion of QED and weak effects into a global PDF fit requires two main modifications. Firstly, hard-scattering matrix elements have to be corrected for EW effects where relevant. This also implies taking into account the contributions from photon-induced subprocesses. This leads to the second main modification, which is the introduction of an additional parton distribution quantifying the photon content of the proton. In turn, this requires generalising the DGLAP evolution equations to account for QED corrections. This generalisation is made possible thanks to the computation of the splitting functions up to $\mathcal{O}(\alpha^2)$ and $\mathcal{O}(\alpha\alpha_s)$ [25,26], with the resulting QED-corrected evolution equations implemented in public PDF evolution codes such as APFEL [27], HOPPET [28], and QEDEVOL [29].

Until recently, two distinct strategies were adopted for the determination the photon PDF: model calculations and data-driven approaches. In the first case, the photon PDF is computed on the base of a theoretically motivated model ansatz. The original realisation of this strategy was the MRST04QED set [21], where the photon PDF was generated at some low scale by one-photon collinear emission off a model for the valence quarks. This led to a simple relation between the photon PDF and the up and down valence distributions, which was then evolved upwards in $Q^2$ using the DGLAP equations corrected for $\mathcal{O}(\alpha)$ contributions.

However, this MRST04QED model accounted only for the *inelastic* component of the photon PDF. In addition to it, one should also account for the *elastic* component, which can be determined by a QED calculation in a model-independent way [30–34] in terms of the electric and magnetic form factors of the proton. This elastic component is derived from the equivalent

photon approximation [35] and accounts for the fact that the proton can emit photons while remaining intact. It was furthermore shown that the elastic component dominates the photon PDF at large-$x$, and that it has associated rather small theoretical uncertainties. In this respect, the CT14QED analysis [24] was originally based on the same ideas as the MRST04QED one, extended with estimate of the uncertainty in their model for the inelastic component based on HERA isolated photon production data [36], but was subsequently complemented with an elastic component following the procedure of [33, 34].

In the second strategy, first advocated by the NNPDF Collaboration, the photon PDF is treated on the same footing as the quark and gluon PDFs. Within this approach, the photon PDF is parametrised in a model-independent way using an artificial neural network and then constrained by LHC Drell-Yan measurements. This procedure was adopted in the NNPDF2.3/3.0QED determinations [23, 37–39]. The limited sensitivity of existing LHC data to PI contributions combined with the use of a flexible parametrisation resulted in large uncertainties on the photon distribution. A similar strategy was adopted in the recent analysis of Ref. [22] in which the ATLAS 8 TeV high-mass Drell-Yan data [40] was employed to constrain the photon PDF. Although this dataset is particularly sensitive to the PI contribution, the resulting photon was still affected by large uncertainties while a reduction in uncertainty is achieved relative to the baseline.

Overcoming the limitations of both two strategies, the LUXqed formalism presented in Refs. [41, 42] represented a breakthrough for the determination of the photon PDF. The LUXqed methodology enhances and introduce corrections to a similar approach adopted by earlier works in Refs. [43–45]. In this framework, both the elastic and the inelastic components of the photon PDF can be expressed in terms of the electromagnetic inclusive structure functions $F_2$ and $F_L$ from lepton-proton scattering by means of an exact QED calculation. This is very advantageous because these structure functions are known rather accurately both experimentally and theoretically. Accounting for the LUXqed constraints then leads to a reduction of the uncertainty of the photon PDF by more than an order of magnitude as compared to the NNPDF3.0QED data-driven determination.

Building upon the recent NNPDF3.1 fit [16], the goal of this paper is to perform a global PDF analysis including QED corrections where the LUXqed calculation is used to constrain the photon PDF. The resulting PDF set, NNPDF3.1luxQED, represents a state-of-the-art determination of the partonic content of the proton including its photon component. The uncertainties on the photon PDF are now at the level of a few percent, with photons carrying up to $\simeq 0.5\%$ of the total proton's momentum.[1] Comparing with NNPDF3.0QED, we find good agreement within uncertainties in the $x \gtrsim 0.02$ region, and larger differences for smaller values of $x$.

We also take a first look at the phenomenological implications of NNPDF3.1luxQED for photon-initiated processes at the LHC. Previous studies based on NNPDF2.3/3.0QED indicated that PI contributions were potentially large, particularly at large invariant masses or transverse momenta, for processes such as Drell-Yan, $W$ pair, and top-quark pair production [47–52]. Indeed, PI effects represented in some cases the dominant source of theoretical uncertainty. We find that photon-initiated corrections computed with NNPDF3.1luxQED can be significant for many processes, leading to corrections of up to 20% depending on the kinematics. These PI contributions are consistent with previous estimates based on NNPDF3.0QED within uncertainties in the kinematic region $Q \gtrsim M_Z$, with larger differences in processes for which $Q < M_Z$.

The outline of this paper is as follows. In Sect. 2 we present the settings of the global NNPDF3.1luxQED analysis in terms of input experimental data, theoretical calculations, and fitting strategy. Then in Sect. 3 we present the NNPDF3.1luxQED set, including a discussion of the momentum fraction of the proton carried by the photon, and in Sect. 4 we discuss some

---

[1]See also [46] for related studies in the MMHT framework.

of its phenomenological implications for PI processes at the LHC. In Sect. 5 we summarise and discuss how our results and the code used to produce them are made publicly available. The full breakdown of the $\chi^2/N_{\text{dat}}$ values in NNPDF3.1luxQED and its comparison with those in NNPDF3.1 is collected in Appendix A.

## 2 Fit settings

In this section we describe the fit configuration of the NNPDF3.1luxQED global analysis. We begin with a review of the input experimental dataset. We then discuss the theoretical framework, including a short summary of the relevant aspects of the LUXqed formalism along with the treatment of QED effects in the DGLAP evolution and the DIS structure functions. Finally, we present the strategy adopted to include the photon PDF in the global fit accounting for the LUXqed theoretical constraints.

### 2.1 Experimental data

The NNPDF3.1luxQED analysis is based on the same dataset as the recent NNPDF3.1 global fit [16]. This dataset includes fixed-target [53–60] and HERA [20] inclusive DIS measurements; charm and bottom cross-sections from HERA [61]; fixed-target Drell-Yan production [62–65]; Tevatron gauge boson and inclusive jet production [66–70]; along with electroweak boson production, inclusive jet, and $t\bar{t}$ cross-sections from ATLAS [71–85], CMS [86–97] and LHCb [98–102]. We refer to Ref. [16] for details about the implementation of each experiment.

For consistency, in this study we use exactly the same dataset as in NNPDF3.1, and in particular the same choice of kinematic cuts. Note that a number of those cuts were determined with the aim of minimising the potential effects from EW corrections and PI contributions. This choice implies that the kinematic regions more sensitive to PI effects are deliberately cut away. In addition, we do not include some recent measurements with known sensitivity to the photon PDF, such as the ATLAS high-mass Drell-Yan measurement at 8 TeV [40], since these were not part of the NNPDF3.1 dataset.

### 2.2 The LUXqed formalism

We briefly review the LUXqed formalism for the determination of the photon PDF, focusing on those features relevant to its implementation in a global analysis. For a comprehensive discussion we refer the reader to Refs. [41,42]. In the LUXqed procedure, the photon PDF can be expressed in terms of the lepton-proton scattering inclusive structure functions $F_2$ and $F_L$ by means of an exact QED calculation as follows:

$$
x\gamma(x,\mu) = \frac{1}{2\pi\alpha(\mu)}\int_x^1 \frac{dz}{z}\left\{\int_{Q^2_{\text{min}}}^{\mu^2/(1-z)}\frac{dQ^2}{Q^2}\alpha^2(Q^2)\left[-z^2 F_L(x/z,Q^2)\right.\right.
$$
$$
\left.\left.+\left(zP_{\gamma q}(z)+\frac{2x^2 m_p^2}{Q^2}\right)F_2(x/z,Q^2)\right]-\alpha^2(\mu)z^2 F_2(x/z,\mu^2)\right\}+\mathcal{O}\left(\alpha\alpha_s,\alpha^2\right),
$$

(2.1)

where $m_p$ is the proton mass, $\mu$ is the factorisation scale, $x$ and $z$ are the momentum fractions, $\alpha$ the running QED coupling, and $P_{\gamma q}$ the photon-quark splitting function. The lower integration limit in the $Q^2$ integral is given by $Q^2_{\text{min}} = (m_p^2 x^2)/(1-z)$.

Note that the integral in $z$ in Eq. (2.1) extends up to $z = 1$. Therefore the LUXqed photon has an explicit dependence upon the elastic component of the structure functions, proportional

to $\delta(1-z)$. This component can be expressed in terms of the electric and magnetic form factors $G_E$ and $G_M$. In [41, 42], the elastic component of $\gamma(x, \mu)$ is determined using the form factors extracted from a fit to world data by the A1 collaboration [103] for $Q^2 \leq 10$ GeV$^2$. The dipole model is then used to extrapolate the form factors to larger values of $Q^2$. A corresponding uncertainty due to the treatment of the large-$Q^2$ extrapolation region is included in the evaluation of the photon PDF.

Furthermore, we observe that the $Q^2$ integral in Eq. (2.1) requires an understanding of the structure functions down to potentially very low scales, well outside the region where perturbative QCD is applicable. Following the prescription of Refs. [41, 42], the integration in $Q^2$ of the inelastic component ($z < 1$) of $F_2$ and $F_L$ is achieved by combining parameterisations of experimental data with the perturbative computation in terms of PDFs where appropriate. Specifically, contributions to the inelastic structure functions come from two regions separated by $W^2 = m_p^2 + Q^2(1-z)/z$. In the resonance region, defined $(m_p + m_\pi)^2 \leq W^2 \leq 3.5$ GeV$^2$, the fit of the CLAS collaboration is used [104]. In order to assess the uncertainty due to this choice, the parametrisation of Ref. [105] is also considered.

The continuum region, defined as $W^2 \geq 3.5$ GeV$^2$, is further subdivided into two regions according the value of $Q^2$. For $Q^2 \leq Q^2_{\text{match}}$, with $Q^2_{\text{match}} = 9$ GeV$^2$, the GD11-P fit by HERMES [106, 107] is employed. For $Q^2 > Q^2_{\text{match}}$, structure functions are computed in terms of PDFs by means of their factorised expressions. The value of $Q^2_{\text{match}}$ can be varied to estimate the uncertainty associated with this particular choice.

As a part of the present work, the LUXqed formalism has been implemented in an open-source public library, FiatLux [108], which has been used to produce the NNPDF3.1luxQED fits. The results obtained with FiatLux have been benchmarked with the original implementation used to produce the results of Refs. [41, 42], finding excellent agreement.

## 2.3   Theoretical calculations

The QCD calculations used in the present analysis are identical to those used in NNPDF3.1 [16]. DGLAP evolution and DIS structure functions are computed at NLO and NNLO accuracy in QCD using APFEL [27]. Heavy-quark mass effects in the structure functions are included using the FONLL general-mass variable-flavour-number scheme [109], and the charm PDF is fitted to data on an equal footing as the light quark and gluon PDFs [110–112]. Hadronic observables are computed at NLO using fast interpolation tables in the APPLgrid [113] and fastNLO [114] formats and combined to the DGLAP evolution kernels using APFELgrid [115]. NNLO corrections to hadronic processes are included by means of point-by-point NNLO/NLO $K$-factors.

In the NNPDF3.1luxQED fit, QCD corrections are supplemented with QED effects. Concerning the evolution of PDFs, on top of the $\mathcal{O}(\alpha)$ corrections, also the $\mathcal{O}(\alpha^2)$ and $\mathcal{O}(\alpha\alpha_s)$ splitting functions computed in Refs. [25, 26] are included in the DGLAP evolution equations. Additionally, $\mathcal{O}(\alpha)$ corrections to the DIS coefficient functions are included. This introduces an additional (but mild) sensitivity to the photon PDF. The implementation in APFEL of the aforementioned QED corrections to the DIS structure functions and to the DGLAP evolution equations, together with the corresponding benchmarking, were presented in Ref. [22]. On the other hand, as in NNPDF3.1, pure weak corrections to hadronic observables are not accounted for.

Using NNPDF3.1luxQED as an input, we find that the cumulative effect on the photon-photon luminosity $\mathcal{L}_{\gamma\gamma}$ of the $\mathcal{O}(\alpha^2)$ and $\mathcal{O}(\alpha\alpha_s)$ corrections to the DGLAP splitting functions ranges between $\simeq 10\%$ at low invariant masses $M_X$ and $\simeq 5\%$ for high $M_X$, see Fig. 2.1. Since these effects are larger than the typical uncertainties on the photon PDF determined through the LUXqed approach, it is important to take them into account. Concerning the DIS structure functions, using NNPDF3.1luxQED, as shown in Fig. 2.1 one finds that the overall impact of

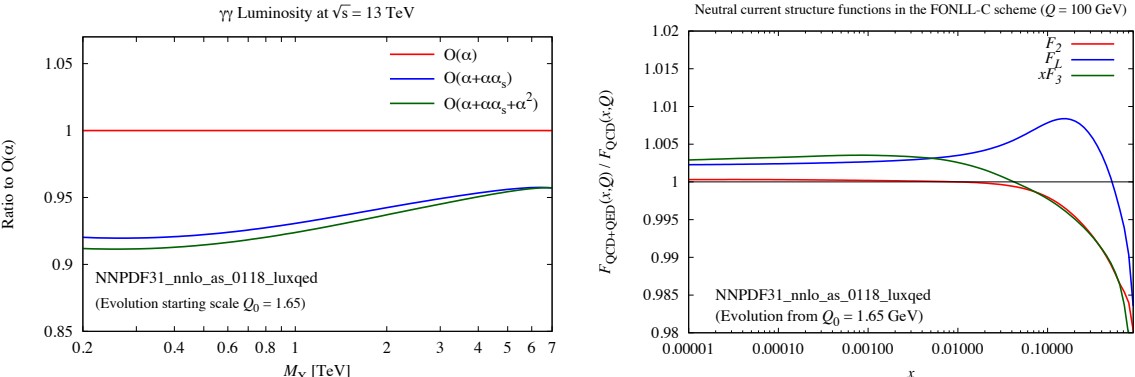

Figure 2.1: Left: comparison of the $\mathcal{L}_{\gamma\gamma}$ luminosity at $\sqrt{s} = 13$ TeV, computed starting from NNPDF3.1luxQED at $Q_0 = 1.65$ GeV and then evolving upwards using different types of QED corrections in the DGLAP splitting functions. Right: comparison between the DIS splitting functions $F_2$, $F_L$, and $xF_3$ at $Q = 100$ GeV with and without QED effects included.

the QED effects is at the permille level, except at large values of $x$ where they can be up to 2%.

In data-driven determinations of the photon PDF, it is in general necessary to include constraints from observables sensitive to PI processes. For instance, in the analysis of Ref. [22] PI contributions to the ATLAS 8 TeV high-mass Drell-Yan data [40] were computed via the aMCfast interface [116] to the Monte Carlo generator MadGraph5_aMC@NLO [117] and used to constrain the photon PDF. In the present study, the photon PDF is determined from a global analysis of hard-scattering data supplemented by the LUXqed theoretical constraint of Eq. (2.1). Given that the NNPDF3.1 dataset and the associated kinematic cuts were specifically designed to minimise the effects of PI contributions, one expects the impact of PI processes to hadronic cross-sections in NNPDF3.1luxQED to be minimal, and thus they are not included here. We have explicitly verified for some of the NNPDF3.1 datasets that this is an excellent approximation, see also the comparisons of Sect. 4. Nevertheless, the approach presented in this work is fully general, and future NNPDF analyses with QED corrections will include collider measurements characterised by sizeable PI contributions.

## 2.4 Fitting strategy

The determination of the NNPDF3.1luxQED set is performed by means of an iterative procedure. The starting point is a prior set of quark and gluon PDFs, in this case NNPDF3.1. From this PDF set, the high-$Q^2$ inelastic component of the photon PDF is computed at $Q = 100$ GeV using Eq. (2.1), while for the other components the same inputs as in Ref. [42] are adopted. The resulting photon PDF is then evolved down to the parametrisation scale $Q_0 = 1.65$ GeV and used as a fixed input in a refit of quark and gluon PDFs.

In this refit, the DGLAP evolution equations consistently include QED effects, the PI contribution to the DIS structure functions is taken into account, and the momentum sum rule reads

$$\int_0^1 dx\, x\, (\Sigma(x, Q_0) + g(x, Q_0) + \gamma(x, Q_0)) = 1\,. \tag{2.2}$$

This procedure, schematically illustrated in Fig. 2.2, is repeated until convergence is reached, in the sense that a stable photon PDF, and thus stable quark and gluon PDFs, are obtained. We consider stable the results of two consecutive iterations where the central value of the photon PDF varies by less than 5% of its uncertainty.

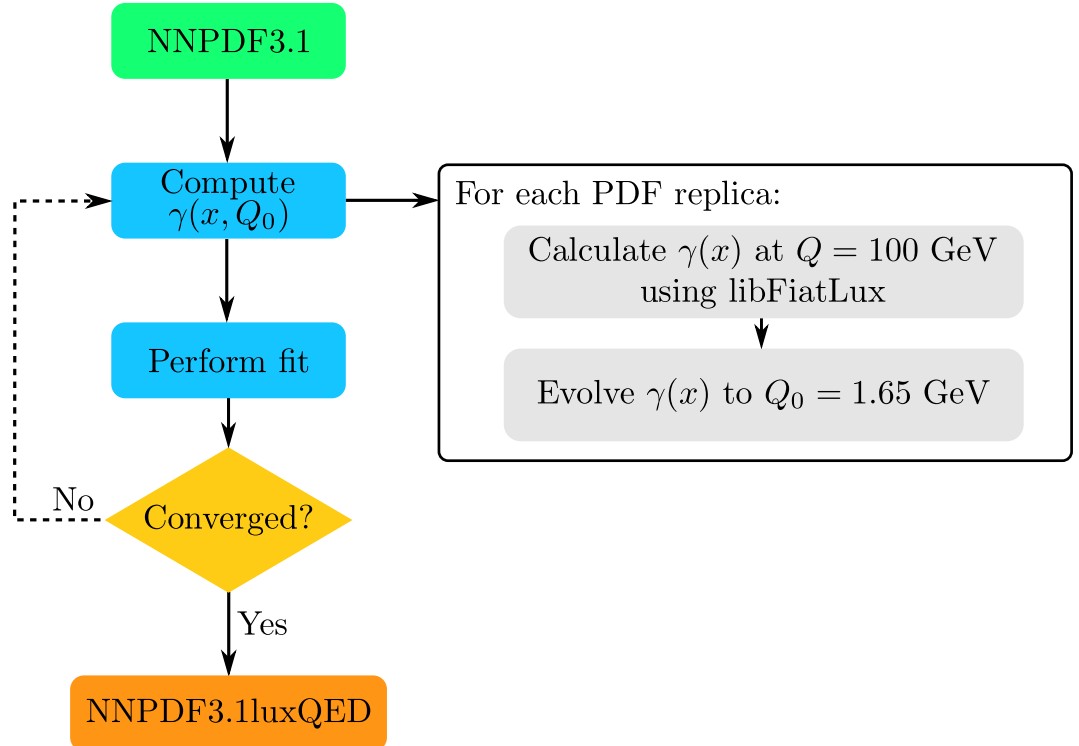

Figure 2.2: Flow diagram representing the NNPDF3.1luxQED fitting strategy. In the last iteration $n_{\text{ite}}$, once the procedure has converged, the additional LUXqed17 are added to $\gamma(x, Q)$, see Sect. 2.5.

As usual in the NNPDF approach, PDF uncertainties are represented by means of an ensemble of $N_{\text{rep}}$ Monte Carlo replicas. Each replica is required to meet a set of quality criteria, discussed in Ref. [38], with fits failing these criteria being discarded. As the present study involves an iterative procedure, one must start with a sample of replicas large enough such that once all $n_{\text{ite}}$ iterations have been completed a significant number of replicas still survives. To this end, here we use a prior sample of $N_{\text{rep}} = 500$ replicas.

Each replica will lead to different high-$Q^2$ DIS structure functions and therefore, by virtue of Eq. (2.1), to a different photon PDF that is used as an external constraint in the following fit. The resulting quark and gluon PDFs are then used as an input to the following iteration of the fitting procedure, until convergence is reached. As the NNPDF3.1 dataset is (by construction) relatively insensitive to the photon PDF, the convergence is rapid and results are stable already after the second iteration. In future analyses, when hadronic measurements sensitive to PI contributions will be included, convergence is likely to be slower.

There are two main differences between our strategy and the direct application of Eq. (2.1) to NNPDF3.1. Firstly, the influence of the photon PDF in the DGLAP evolution equations and in the DIS structure functions is consistently taken into account during the fit of the quark and gluon PDFs. Secondly, the contribution of the photon PDF to the total momentum fraction is properly treated by imposing the momentum sum rule Eq. (2.2) during the fits. While these effects are likely to be small in this specific analysis, our framework is fully general and allows for the consistent inclusion of hadronic observables sensitive to the photon-initiated contributions.

In order to illustrate the convergence of the procedure, in Fig. 2.3 we show a comparison of the photon, gluon, up quark, and down quark PDFs at $Q = 100$ GeV between the first and the second iteration (labelled ITE1 and ITE2, respectively). We have verified that

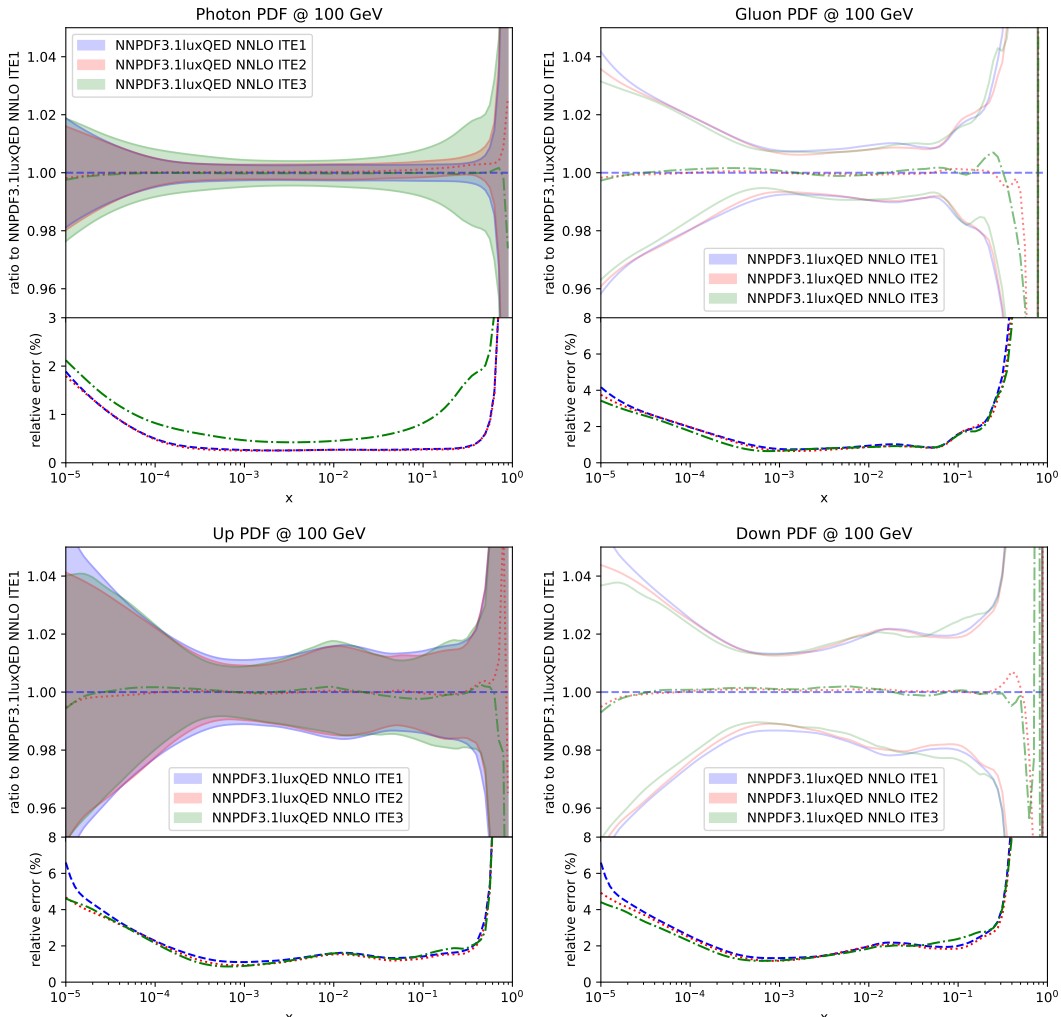

Figure 2.3: Comparison of the photon, gluon, up quark, and down quark PDFs at $Q = 100$ GeV between the first and second iterations fit (ITE1 and ITE2 respectively) of the fitting procedure sketched in Fig. 2.2. For completeness, we also show the third and final iteration of the procedure $n_{\text{ite}}$ (ITE3) where the additional LUXqed17 systematic variations have been added to $\gamma(x, Q)$, see Sect. 2.5

additional iterations leave the photon PDF unchanged, demonstrating that stability has been reached. For completeness, in Fig. 2.3 we also show the third and final iteration of the procedure (ITE3), where the additional LUXqed17 systematic variations are added to the photon PDF (see Sect. 2.5). As expected, these have the largest impact in the region $x \gtrsim 0.05$, where the elastic contribution to the photon PDF is most important.

## 2.5 The uncertainties on the photon PDF

As mentioned above, the calculation of the photon PDF in terms of structure functions involves several contributions: the elastic component, the inelastic resonance component, and the inelastic low- and high-$Q^2$ continuum components. Only the last component can be factorised in terms of PDFs and perturbative coefficient functions. Therefore, the ensemble of $N_{\text{rep}}$ Monte Carlo replicas of the photon PDF accounts only for a part of the uncertainty, namely the one associated to the inelastic high-$Q^2$ component. For a comprehensive estimate of the uncertainty

one must also account for a number of additional sources of error.

The following sources of uncertainty are considered [42]: the elastic contribution from the A1 world proton form factor fits [103]; the parametrisation of the DIS structure functions in the resonance region [104–106]; the parametrisation of $R_{L/T}$ [107, 118, 119], the ratio between longitudinal and transverse structure functions; the scale $Q^2_{\text{match}}$ at which low- and high-$Q^2$ inelastic structure functions are matched; a twist-4 modification of the longitudinal structure function $F_L$ [120, 121]; and finally an estimate of the missing higher-order corrections in the calculation of the DIS structure functions at high $Q^2$.

In the NNPDF3.1luxQED analysis, these uncertainties are introduced at the last iteration of the procedure. Once the quark and gluon PDFs from the $(n_{\text{ite}} - 1)$-th iteration have been determined, they are used to construct $\gamma_{n_{\text{ite}}}(x, Q)$. Then, for each photon PDF replica of this last iteration, $\gamma^{(k)}_{n_{\text{ite}}}$, $n_{\text{sys}} = 7$ extra uncertainties are included as statistical fluctuations upon the photon PDF at $Q = 100$ GeV with correlations in $x$, namely:

$$\widetilde{\gamma}^{(k)}_{n_{\text{ite}}}(x, Q) = \gamma^{(k)}_{n_{\text{ite}}}(x, Q) + \sum_{j=1}^{n_{\text{sys}}} \delta\gamma^{(\text{lux})}_j(x, Q) \cdot \mathcal{N}(0, 1), \quad k = 1, \dots, N_{\text{rep}}, \qquad (2.3)$$

where $\mathcal{N}(0, 1)$ is an univariate Gaussian random number and $\delta\gamma^{(\text{lux})}_j$ is the normalised eigenvector for the $j$-th systematic uncertainty in LUXqed17. Specifically, $\delta\gamma^{(\text{lux})}_j$ is obtained through the diagonalisation of the covariance matrix for the extra LUXqed17 uncertainties defined on a grid of $x$ points, using a similar method as that of Refs. [122, 123]. We have verified that this approach is numerically equivalent to using the corresponding Hessian eigenvectors of LUXqed17.

The photon PDF defined in Eq. (2.3) is finally used as an input for the $n_{\text{ite}}$-th fit iteration, to determine the final set of quark and gluons of NNPDF3.1luxQED.

# 3 The NNPDF3.1luxQED set

In this section we discuss the results of the iterative procedure outlined in Sect. 2.4: the NNPDF3.1luxQED fit. We focus on the NNLO case and comment where appropriate on any differences with respect to NLO. The overall fit quality of NNPDF3.1luxQED is $\chi^2/N_{\text{dat}} = 1.168$ at NLO and $\chi^2/N_{\text{dat}} = 1.148$ at NNLO. While there is some variation dataset-to-dataset, the global fit quality is identical to the corresponding NNPDF3.1 results at NNLO. See Appendix A for a full breakdown of the data description at NNLO and its comparison with NNPDF3.1.

## 3.1 The photon PDF

Here we compare our results for the photon PDF $\gamma(x, Q)$ with those of the NNPDF3.0QED and LUXqed16/17 PDF sets. Comparisons with the latter are performed always at $Q \geq 10$ GeV, as the LUXqed16/17 sets are not defined below this scale. As discussed in Sect. 9.2 of Ref. [42], the LUXqed17 set has a improved evaluation of the photon PDF calculation and of the associated error estimates in comparison to LUXqed16. In the left panel of Fig. 3.1 we compare the NNPDF3.1luxQED photon PDF at $Q = 100$ GeV with the corresponding results from LUXqed16 and LUXqed17, normalised to the central value of the latter. The three determinations agree well across the full $x$ range, with central values always compatible within uncertainties. In addition, for $x \gtrsim 0.1$ the total uncertainties on the photon PDF from NNPDF3.1luxQED and LUXqed16/17 are identical. This feature is explained by the fact that in this region the uncertainties due to the elastic and low-$Q^2$ inelastic structure functions dominate.

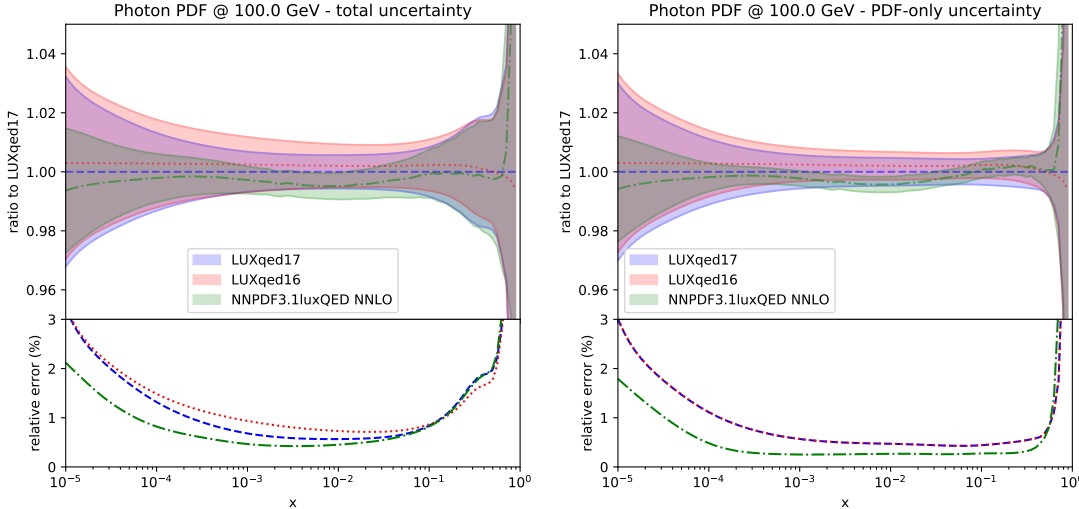

Figure 3.1: Left: comparison of the NNPDF3.1luxQED photon at $Q = 100$ GeV with that of LUXqed16/17 normalized to the central value of the latter. The bottom panel indicates the relative uncertainty on the photon PDF in each case. Right: the same comparison, now including only the uncertainties on $\gamma(x, Q)$ related to the quark and gluon PDFs in the high-$Q^2$ inelastic component.

At medium- and small-$x$, the NNPDF3.1luxQED photon exhibits somewhat smaller uncertainties. This is due to the use of a different set of quark and gluon PDFs determining the high-$Q^2$ inelastic component, specifically NNPDF3.1 rather than the PDF4LHC15 set [124] used in LUXqed16/17. The contribution from the different error sources is further illustrated in the right panel of Fig. 3.1 where the same comparison including only the uncertainties due to the high-$Q^2$ inelastic component is shown. The plot shows how at medium- to small-$x$ the contribution from the high-$Q^2$ inelastic structure functions dominates the overall uncertainty.

In order to gauge the stability of the photon PDF with respect to the perturbative order of the QCD calculations used in the fit, in Fig. 3.2 we compare the photon PDFs from the NNPDF3.1luxQED NLO and NNLO fits, normalised to the central value of the former. The photon distributions are consistent within uncertainties, demonstrating good perturbative stability. Indeed, the shift due to the change in perturbative order is outside the PDF error bands only in the small-$x$ region, where the photon is sensitive to the prior PDF used for the computation of the high-$Q^2$ inelastic component. In addition, we find that the photon PDF uncertainties are unaffected by the variation of the perturbative order.

In order to quantify the differences between photon PDFs determined from global analyses with and without imposing the LUXqed theoretical constraint, we compare NNPDF3.1luxQED with NNPDF3.0QED. In the following, the PDF uncertainties of NNPDF3.0QED are computed as 68% confidence-level (CL) intervals, with the central value taken to be the midpoint of the interval. In Fig. 3.3 we show the photon distributions from these two sets at $Q = 1.65$ GeV (left plot) and $Q = 100$ GeV (right plot). We find that both at low and high scales, in the region $x \gtrsim 2 \times 10^{-2}$ the two determinations agree within uncertainties. For $x \lesssim 2 \times 10^{-2}$ instead, the NNPDF3.0QED photon undershoots NNPDF3.1luxQED by up to 40% at $Q = 100$ GeV. At high scales, PDF uncertainties in NNPDF3.0QED are at the level of a few percent at small $x$ but become as large as almost 100% at large $x$. The uncertainties in NNPDF3.1luxQED are instead at the level of a few percent over the entire range in $x$ (see also Fig. 3.1).

As shown in Fig. 3.3, for $x \lesssim 10^{-2}$ the NNPDF3.0QED photon undershoots the 3.1luxQED one both at low and at high scales by an amount which is not covered by the PDF uncertainties

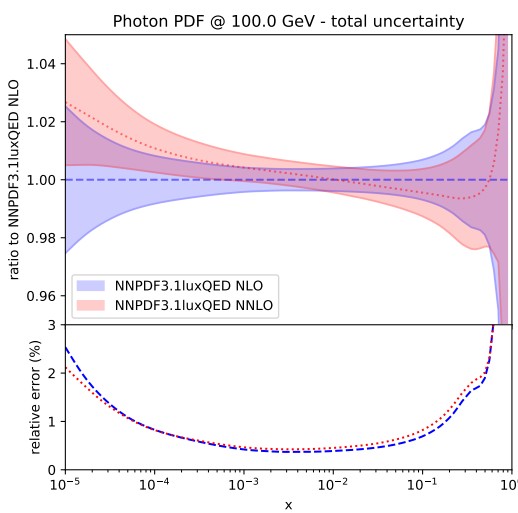

Figure 3.2: Comparison between $\gamma(x,Q)$ in the NNPDF3.1luxQED NLO and NNLO fits.

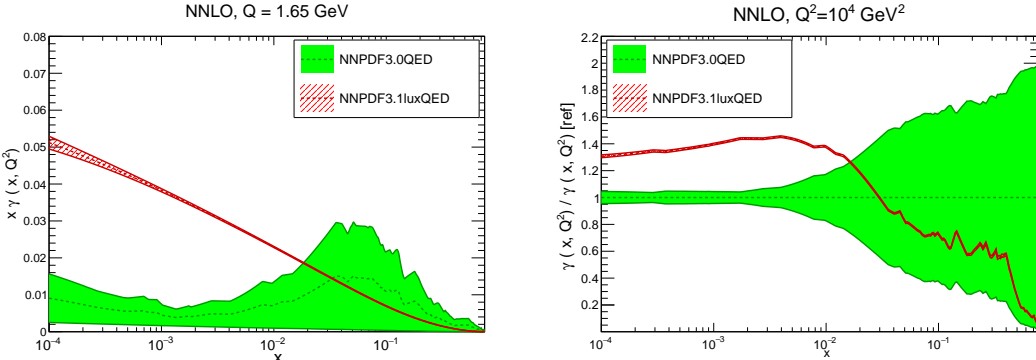

Figure 3.3: Comparison between the photon PDF $\gamma(x,Q)$ in NNPDF3.0QED and in NNPDF3.1luxQED at $Q = 1.65$ GeV (left) and at $Q = 100$ GeV (right plot). In the latter case, results are normalised to the central value of NNPDF3.0QED.

of the former. There are at least two possible contributions to such differences. First of all, the inclusion of $\mathcal{O}(\alpha^2)$ and $\mathcal{O}(\alpha\alpha_s)$ terms in the DGLAP equations (absent in NNPDF3.0QED), accounts to up to a difference of 5% when the photon PDF is evolved from $Q_0 = 1.65$ GeV to $Q = 100$ GeV (see also Fig. 2.1), explaining part of the discrepancy.

The second, and more important, potential reason is related to the fact that in NNPDF2.3QED the boundary condition $\gamma(x,Q_0)$ was determined from a fit to DIS and Drell-Yan cross-sections using different settings for the QCD+QED evolution equations [39] as compared to those used later to construct NNPDF3.0QED. This partial mismatch then seems to lead to a suppression of the photon PDF at small-$x$, explaining some of the differences observed in Fig. 3.3. In this context, recall than in NNPDF2.3QED the photon PDF was constrained mostly by the LHC Drell-Yan measurements, which makes tricky the mapping between how different evolution settings translate into a change in the fitted boundary condition. In any case, is clear that pinning down the underlying origin of these differences for $x \lesssim 10^{-2}$ would require redoing the NNPDF2.3QED fit with exactly the same theoretical settings for the DGLAP evolution as in NNPDF3.1QED, which is beyond the scope of this paper.

It is also interesting to examine the relative size of the photon PDF with respect to the

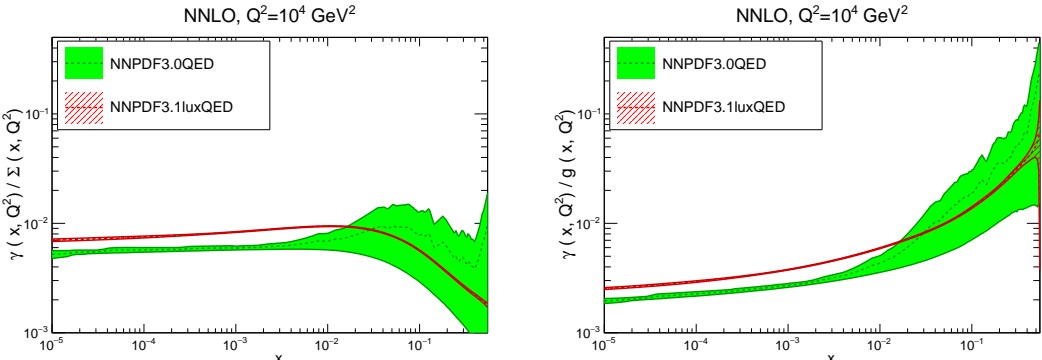

Figure 3.4: The ratios of the photon PDF to the quark singlet $\gamma(x,Q)/\Sigma(x,Q)$ (left) and to the gluon $\gamma(x,Q)/g(x,Q)$ (right) PDFs for $Q = 100$ GeV, comparing NNPDF3.0QED and NNPDF3.1luxQED. The corresponding LUXqed17 results are very similar to the NNPDF3.1luxQED ones and thus not shown.

total quark singlet and gluon PDFs. This allows us to estimate where PI contribution to hadron-collider processes becomes sizeable as compared to quark- and gluon-initiated sub-processes. In Fig. 3.4 we compare the predictions of NNPDF3.0QED and NNPDF3.1luxQED for the $\gamma(x,Q)/\Sigma(x,Q)$ (left) and $\gamma(x,Q)/g(x,Q)$ (right) ratios at $Q = 100$ GeV. From the right panel of Fig. 3.4 we observe that the $\gamma/\Sigma$ ratio is around $\mathcal{O}(10^{-2}) \simeq \mathcal{O}(\alpha_{\mathrm{QED}})$ over the entire range of $x$. On the other hand, from the right panel of Fig. 3.4 we find that for $x \gtrsim 0.01$ the $\gamma/g$ ratio becomes larger than the $\gamma/\Sigma$ one. In fact, the $\gamma/g$ ratio is as large as 10% for $x \simeq 0.5$.

For completeness, we find that that complementing a data-driven determination of the photon PDF with the LUXqed theoretical constraints allows for a precise determination of the photon PDF in most of the range of $x$ relevant for applications at the LHC.

## 3.2  QED effects on the quark and gluon PDFs

In this section we study the quark and gluon PDFs in NNPDF3.1luxQED as compared to their corresponding QCD-only counterparts in NNPDF3.1. This comparison gauges the impact on quarks and gluons of three different QED effects: the modification of the momentum sum rule, the QED splitting functions in the DGLAP evolution equations, and the QED corrections to the DIS coefficient functions.

In Fig. 3.5 we show the singlet and gluon PDFs of the NNPDF3.1 and NNPDF3.1luxQED sets at $Q = 100$ GeV normalised to the central value of the former. While differences at the level of the singlet are small, differences for the gluon PDF are somewhat larger. Indeed, the NNPDF3.1luxQED gluon is smaller than its QCD counterpart by about 1% at $x \simeq 10^{-2}$ and enhanced by about 5% for $x \simeq 0.5$. In both cases, the shift in the central values is at the edge of the corresponding PDF uncertainty band. The effect on the gluon PDF can be explained by observing that, as we will discuss in Sect. 3.4, the photon PDF can carry up to 0.5% of the proton momentum. This fraction is effectively subtracted from the singlet and gluon distributions by means of the sum rule, Eq. (2.2). However, the sum rule mostly affects the gluon PDF because the normalisation of the quark singlet is more tightly constrained from the DIS inclusive structure function data. We conclude that the back-reaction of QED effects onto the quark and gluon PDFs is small but not negligible, particularly for the latter.

For completeness, in Fig. 3.6 we show the same comparison as in Fig. 3.5 but now between NNPDF3.1luxQED and LUXqed17. Note that the quark and gluon PDFs of LUXqed17 correspond closely to those of the PDF4LHC15 set, differing only by a rescaling of the gluon PDF

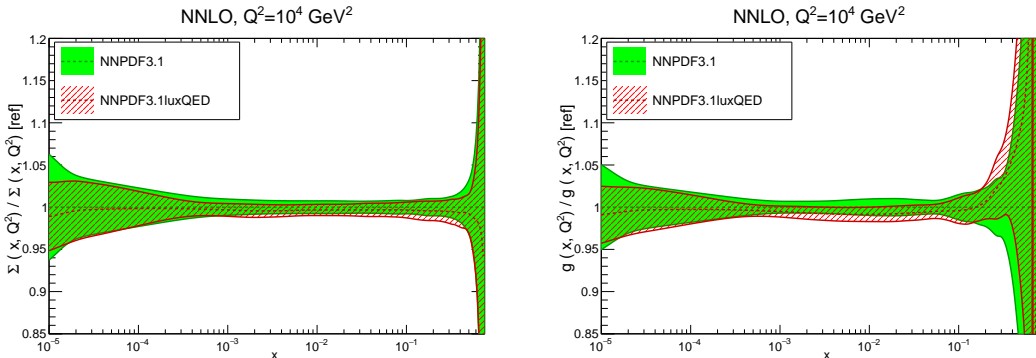

Figure 3.5: Comparison of the total quark singlet (left) and gluon PDFs (right) between NNPDF3.1 and NNPDF3.1luxQED at $Q = 100$ GeV, normalised to the central value of the former.

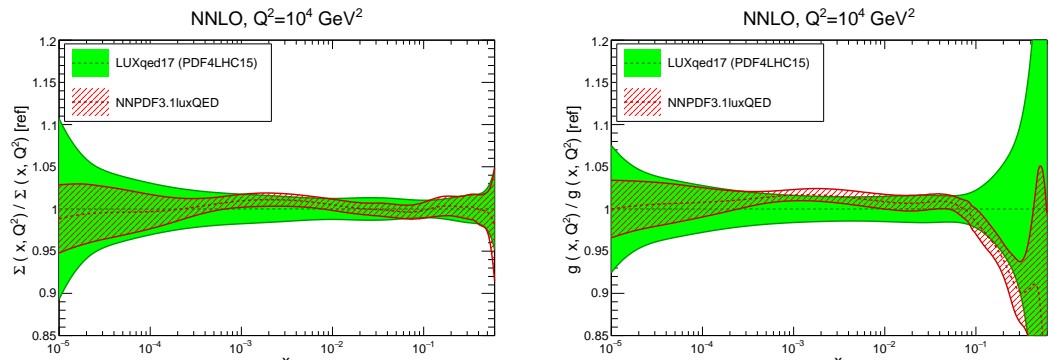

Figure 3.6: Same as Fig. 3.5, now comparing the quark singlet and gluon of NNPDF3.1luxQED with those of LUXqed17, which correspond closely to the PDF4LHC15 NNLO set.

and by the QED contributions to the DGLAP evolution. Since the PDF4LHC15 set is built as a combination of three different PDF sets, namely CT14, MMHT14, and NNPDF3.0, it exhibits larger uncertainties that the individual sets. We find good compatibility between the singlet and gluon of NNPDF3.1luxQED and LUXqed17, with the PDF errors of the former being rather smaller. These reduced uncertainties are particularly noticeable for the medium and large-$x$ gluon PDF, due to the several gluon-sensitive experiments included in NNPDF3.1 such as top-quark pair distributions [125] and the $Z$ boson $p_T$ [126].

### 3.3 Partonic luminosities

Next we compare partonic luminosities integrated over rapidity as a function of the final-state invariant mass $M_X$ for photon-photon and photon-quark initial states (see [48] for the definitions used). In Fig. 3.7 we compare the $\mathcal{L}_{\gamma\gamma}$ luminosity obtained with the NNPDF3.0QED, NNPDF3.1luxQED, and LUXqed17 sets for a centre-of-mass energy of $\sqrt{s} = 13$ TeV. From the left panel of Fig. 3.7 we observe good agreement between NNPDF3.0QED and NNPDF3.1luxQED. The two sets agree within uncertainties over the entire mass range considered, except for $\mathcal{L}_{\gamma\gamma}$ at $M_X \lesssim 30$ GeV where NNPDF3.1luxQED overshoots NNPDF3.0QED.

Evident once again is the effect of the LUXqed constraints on the photon PDF uncertainty, with the errors of just a few percent as compared to the determination that does not account

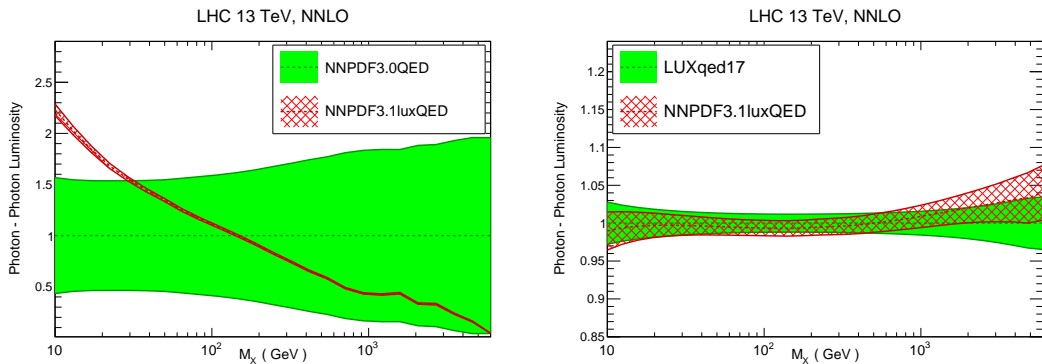

Figure 3.7: Comparison of the $\gamma\gamma$ PDF luminosities between NNPDF3.1luxQED and NNPDF3.0QED (left) and LUXqed17 (right plot) as a function of $M_X$ for $\sqrt{s} = 13$ TeV. Note that the $y$ axis range is different in the both plots.

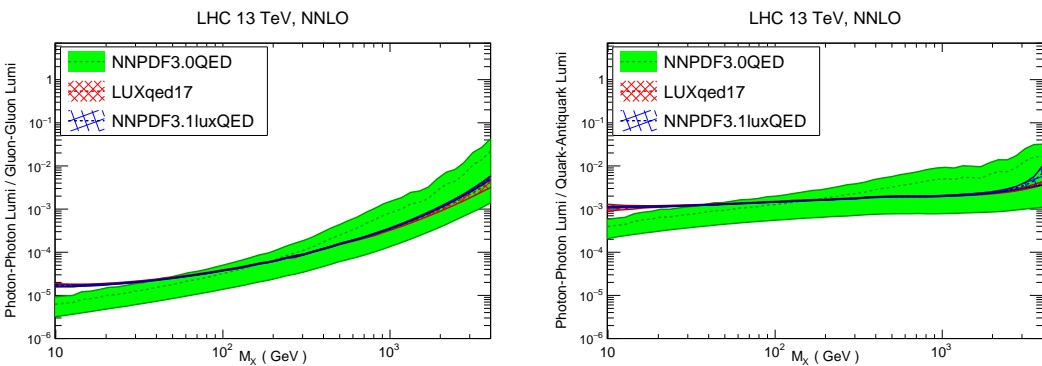

Figure 3.8: The $\mathcal{L}_{\gamma\gamma}/\mathcal{L}_{gg}$ (left) and $\mathcal{L}_{\gamma\gamma}/\mathcal{L}_{q\bar{q}}$ (right plot) ratios of PDF luminosities.

for them. From Fig. 3.7 we also see that there is good agreement between LUXqed17 and NNPDF3.1luxQED both at the level of central values and of uncertainties. Only at $M_X \gtrsim 1$ TeV NNPDF3.1luxQED tends to be a few percent larger than LUXqed17. Similar considerations hold for the $\mathcal{L}_{q\gamma}$ luminosities.

Following the PDF-level comparisons presented in Sect. 3.1, it is instructive to also consider the ratios of the photon-photon luminosity over gluon-gluon and over quark-antiquark luminosities, $\mathcal{L}_{\gamma\gamma}/\mathcal{L}_{gg}$ and $\mathcal{L}_{\gamma\gamma}/\mathcal{L}_{q\bar{q}}$. These ratios are interesting since they provide an estimate of the relative importance of the PI contribution over quark- and gluon-initiated contributions as a function of $M_X$. These ratios are shown in Fig. 3.8, where we compare the results from NNPDF3.0QED, NNPDF3.1luxQED, and LUXqed17. The $\mathcal{L}_{\gamma\gamma}/\mathcal{L}_{q\bar{q}}$ ratio varies very mildly with $M_X$, with a value $\simeq 10^{-3}$. On the other hand, the ratio $\mathcal{L}_{\gamma\gamma}/\mathcal{L}_{gg}$ increases steeply with the final state invariant mass $M_X$, beginning at $\simeq 10^{-5}$ at low invariant masses and growing up to $\simeq 10^{-3}$ for $M_X = 4$ TeV. For large invariant masses, we therefore find that the two ratios $\mathcal{L}_{\gamma\gamma}/\mathcal{L}_{gg}$ and $\mathcal{L}_{\gamma\gamma}/\mathcal{L}_{q\bar{q}}$ take similar values. For both cases, the results of the three PDF sets are consistent within uncertainties, with NNPDF3.1luxQED and LUXqed17 exhibiting significantly reduced errors as compared to NNPDF3.0QED.

## 3.4 The momentum fraction carried by photons

As the photon PDF carries a non-zero amount of the total proton momentum, it therefore contributes to the momentum sum rule, Eq. (2.2). Here we examine the momentum fraction

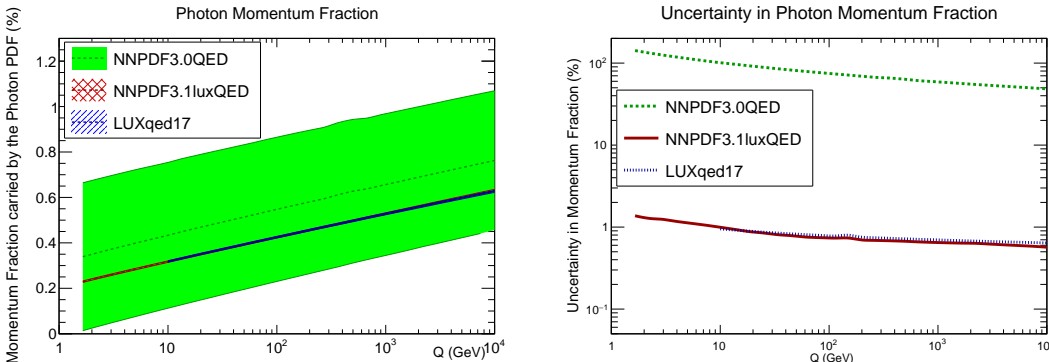

Figure 3.9: The momentum fraction $\langle x \rangle_\gamma$ carried by photons in the proton (left) and its percentage uncertainty (right) as a function of $Q$ for NNPDF3.0QED, NNPDF3.1luxQED, and LUXqed17.

|  | $\langle x \rangle_\gamma \, (Q = 1.65 \, \text{GeV})$ | $\langle x \rangle_\gamma \, (Q = m_Z)$ |
|---|:---:|:---:|
| NNPDF3.0QED | $(0.3 \pm 0.3)\%$ | $(0.5 \pm 0.3)\%$ |
| NNPDF3.1luxQED | $(0.229 \pm 0.003)\%$ | $(0.420 \pm 0.003)\%$ |
| LUXqed17 | – | $(0.421 \pm 0.003)\%$ |

Table 3.1: The momentum fraction $\langle x \rangle_\gamma$ carried by photons in the proton, Eq. (3.1), at the initial parametrization scale $Q = Q_0 = 1.65$ GeV and at typical LHC scale $Q = m_Z$.

carried by the photon PDF

$$\langle x \rangle_\gamma (Q) \equiv \int_0^1 dx \, x\gamma(x, Q) \,. \tag{3.1}$$

In Fig. 3.9 we show the value of $\langle x \rangle_\gamma$ in the NNPDF3.1luxQED, NNPDF3.0QED, and LUXqed17 sets as a function of the scale $Q$. In the right plot of Fig. 3.9 we also show the corresponding percent PDF uncertainties. For LUXqed17 we restrict the comparison to the validity region of this set, *i.e.* $Q \geq 10$ GeV.

From Fig. 3.9 we observe that, while the NNPDF3.0QED determination is affected by large PDF uncertainties, the other two sets lead to a compatible prediction for the photon momentum fraction for all values of $Q$. As expected from the PDF-level comparisons, there is a significant reduction in the uncertainty on the value of $\langle x \rangle_\gamma$ once the LUXqed theoretical constraints are accounted for. Indeed, while in NNPDF3.0QED the uncertainties in the photon momentum fraction range from around 50% to 100%, in NNPDF3.1luxQED the contribution of the photon PDF to the momentum of the proton is known with an accuracy better than 1% over the entire range in $Q$. Nevertheless, the central value of $\langle x \rangle_\gamma$ in NNPDF3.0QED turns out to be rather close to that of NNPDF3.1luxQED, highlighting the consistency between the two approaches.

In Tab. 3.1 we report the photon momentum fraction Eq. (3.1) both at the initial parametrisation scale $Q_0 = 1.65$ GeV and at $Q = m_Z$ for the three PDF sets including the associated uncertainties. While in NNPDF3.0QED the photon momentum fraction at the initial scale is consistent with zero, in NNPDF3.1luxQED one finds a non-zero photon momentum fraction with very high statistical significance. In particular, the photon momentum fraction in NNPDF3.1luxQED increases from 0.23% at low scales to 0.42% at $Q = m_Z$, with small uncertainties in both cases. For $Q = m_Z$, the results of NNPDF3.1luxQED are fully consistent with those of LUXqed17, as also shown in Fig. 3.9.

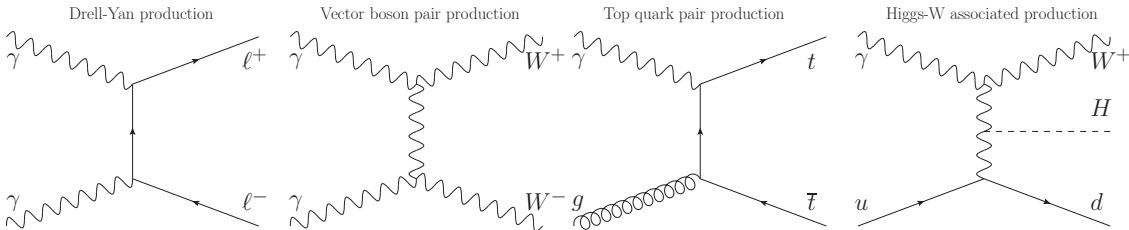

Figure 4.1: Representative PI diagrams for various LHC processes: Drell-Yan, vector-boson pair production, top-quark pair production, and the associated production of a Higgs with a $W$ boson.

# 4 Photon-initiated processes at the LHC

We shall now explore some of the implications of NNPDF3.1luxQED for LHC phenomenology. Specifically, we shall investigate the application of this new set to the study of Drell-Yan, vector-boson pair production, top-quark pair production, and the associated production of a Higgs boson with a $W$ boson. Representative PI diagrams contributing to these processes at the Born level are shown in Fig. 4.1. Our aim is to assess the relative size of the PI contributions with respect to quark- and gluon-initiated subprocesses at the $\sqrt{s} = 13$ TeV LHC. See also Refs. [24, 33, 47–51] for recent studies.

The results presented in this section have been obtained at leading order in both the QCD and QED couplings using `MadGraph5_aMC@NLO` interfaced to `APPLgrid` through `aMCfast`. We have used the default values in `MadGraph5_aMC@NLO` v2.6.0 for the couplings and other electroweak parameters, as defined in the standard model setup. In particular, we use the default value $\alpha = 1/132.51$ for the QED coupling and ignore the effects of its running that are beyond the accuracy of the calculation.

We will compare the predictions of NNPDF3.1luxQED to those of NNPDF3.0QED and LUXqed17. In all cases we will use the NNLO PDF sets, though the photon PDF depends only mildly on the perturbative order (see Fig. 3.2). PDF uncertainties for the NNPDF sets are defined as the 68% confidence level interval and the central value as the midpoint of this range. This is particularly relevant for NNPDF3.0QED for which, due to non-Gaussianity in the replica distribution, PDF errors computed as standard deviations can differ by up to one order of magnitude as compared to the 68% CL intervals.

## 4.1 Drell-Yan production

We begin by examining the role of PI contributions in neutral-current Drell-Yan production. We will study this process in three different kinematic regions of the outgoing lepton pair: around the $Z$ peak, at low invariant masses, and at high invariant masses. We start with the $Z$ peak region, defined as $60 \leq M_{ll} \leq 120$ GeV, where $M_{ll}$ is the lepton-pair invariant mass, and focus on the central rapidity region $|y_{ll}| \leq 2.5$, relevant for ATLAS and CMS.[2] This region provides the bulk of the Drell-Yan measurements included in modern PDF fits and therefore assessing the impact of PI contributions is particularly important here.

In Fig. 4.2 we show the ratio of the PI contributions to the corresponding quark- and gluon-initiated contributions for Drell-Yan production as a function of $M_{ll}$ at $\sqrt{s} = 13$ TeV in the $Z$ peak region. We compare the predictions of NNPDF3.0QED, LUXqed17, and NNPDF3.1luxQED, with the PI contributions normalised to the central value of

---

[2]We have verified that similar results hold for the forward rapidity region, $2.0 \leq y_{ll} \leq 4.5$, relevant for LHCb.

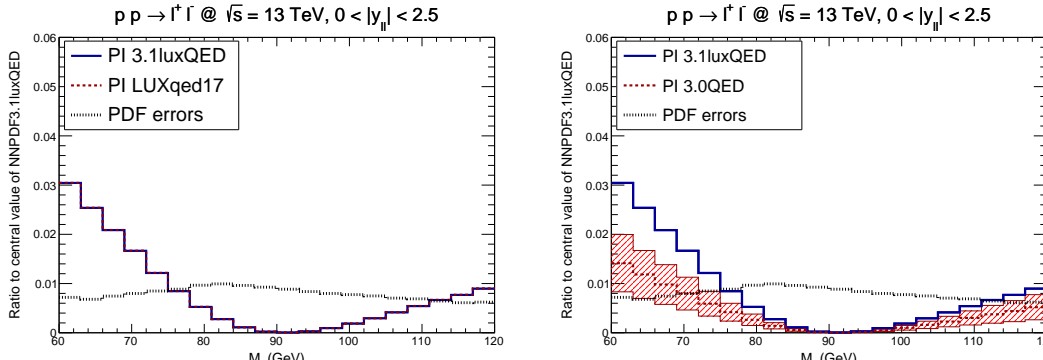

Figure 4.2: The ratio of photon-initiated contributions to the corresponding quark- and gluon-initiated ones for neutral current Drell-Yan production as function of the lepton-pair invariant mass $M_{ll}$ in the $Z$ peak region and central rapidities $|y_{ll}| \leq 2.5$ at $\sqrt{s} = 13$ TeV. We compare NNPDF3.0QED, LUXqed17, and NNPDF3.1luxQED, with the PI contributions in each case normalized to the central value of the latter. The NNPDF3.0QED uncertainty band is represented by the red band. For reference, we also indicate the value of the PDF uncertainties in NNPDF3.1luxQED.

NNPDF3.1luxQED. For reference we also show the value of the PDF uncertainties in NNPDF3.1luxQED.

We find that PI effects for this process are at the permille level for $M_{ll} \sim M_Z$ but they become larger as we move away from the $Z$ peak, reaching 3% for $M_{ll} = 60$ GeV. At the lower edge of the $M_{ll}$ region the contribution of the PI channel exceeds the level of PDF uncertainty, highlighting the sensitivity of this distribution to the photon PDF. We find that NNPDF3.1luxQED and LUXqed17 lead to a larger PI contribution as compared to NNPDF3.0QED at low $M_{ll}$. As the PI contribution is only significant away from the $Z$-peak, where the bulk of the cross-section lies, these effects may be reasonably neglected in the integrated cross-sections.

Fig. 4.2 demonstrates that the PI contributions in NNPDF3.1luxQED and LUXqed17 lead to very similar results for Drell-Yan production around the $Z$ peak. We have verified that this similarity holds also for the low and high mass kinematic regions, as well as for the rest of processes studied in this section. In the following discussion we will therefore restrict ourselves to comparisons between NNPDF3.0 and NNPDF3.1luxQED.

We now move to study the low- and high-mass regions, defined as $15 \leq M_{ll} \leq 60$ GeV and $M_{ll} \geq 400$ GeV respectively. Drell-Yan low-mass measurements have been presented by ATLAS, CMS, and LHCb [84,127,128], with the two-fold motivation of providing input for PDF fits and to study QCD in complementary kinematic regimes. The high-mass region is relevant for BSM searches that exploit lepton-pair final states, such as those expected in the presences of new heavy gauge bosons $W'$ or $Z'$ [129,130].

In Fig. 4.3 we show the same comparison as in the right panel of Fig. 4.2 for the low- and high-mass regions. In the low-mass case, the PI effects are more significant than in the $Z$-peak region, being between 3% and 4% for most of the $M_{ll}$ range, consistently larger that the PDF uncertainty. We find that PI effects in NNPDF3.1luxQED can be up to a factor three larger than in the NNPDF3.0QED case due to the corresponding differences in the photon PDF at small $x$. In the case of the high-mass region, we observe that the effect of the PI contribution computed with NNPDF3.1luxQED is comparable to the PDF uncertainties for $M_{ll} \gtrsim 3$ TeV, eventually becoming as large as $\simeq 10\%$ of the QCD cross-section. These effects are markedly smaller than in NNPDF3.0QED, where shifts in the cross-section up to $\simeq 80\%$ due to PI contributions were allowed within uncertainties.

To conclude this discussion on Drell-Yan at the LHC, we have evaluated the ratio of the

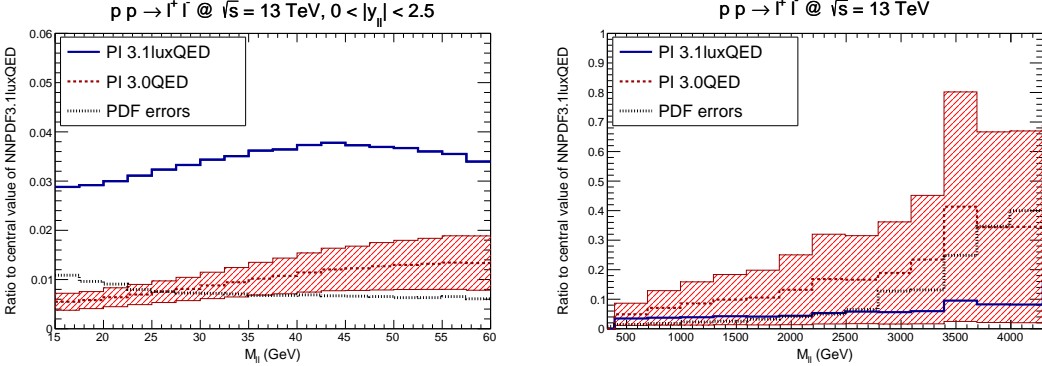

Figure 4.3: Similar representation as the right panel of Fig. 4.2 for the low (left) and high (right plot) invariant mass regions, defined as 15 GeV $\leq M_{ll} \leq$ 60 GeV and $M_{ll} \geq$ 400 GeV respectively. Please note that the right figure is plotted in a larger $y$-axis range in comparison to previous plots.

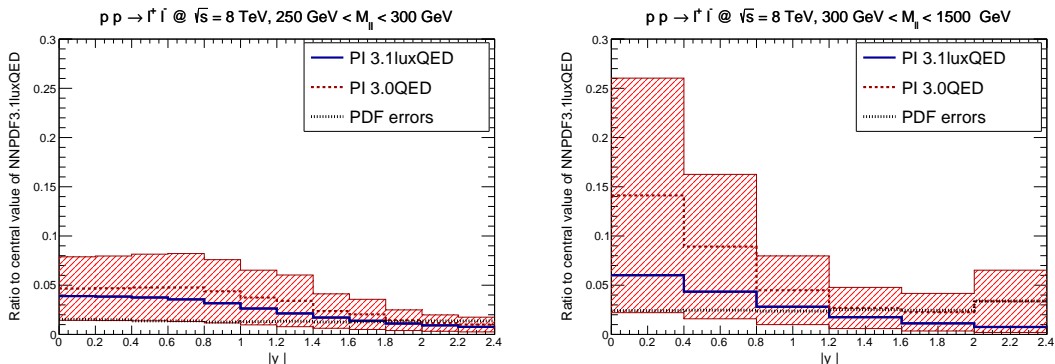

Figure 4.4: Same as Fig. 4.3 for the kinematics of the ATLAS high-mass DY measurement at 8 TeV. We show results for the lepton-pair rapidity distributions $|y_{ll}|$ in two different invariant mass bins, 250 GeV $\leq M_{ll} \leq$ 300 GeV (upper) and 300 GeV $\leq M_{ll} \leq$ 1500 GeV (lower plots).

LO PI contributions to the NLO QCD cross-sections for the kinematics of the ATLAS high-mass Drell-Yan measurements at 8 TeV [40]. Both the Bayesian reweighting study of the ATLAS paper [40] and the analysis of Ref. [22] indicate that this dataset has a considerable sensitivity to PI contributions if NNPDF3.0QED is used as a prior. Here we revisit this process to assess how the picture changes when using NNPDF3.1luxQED.

In Fig. 4.4 we show the ratio of PI over QCD contributions for the lepton-pair rapidity distributions $|y_{ll}|$ in Drell-Yan at 8 TeV for two invariant mass bins, 250 GeV $\leq |M_{ll}| \leq$ 300 GeV and 300 GeV $\leq |M_{ll}| \leq$ 1500 GeV. As can be seen, with NNPDF3.0QED the effects of the PI contribution at large invariant masses can be as large as 25% of the QCD cross-section. This shift is larger than the corresponding experimental uncertainties, which are typically at the percent level, explaining the sensitivity of NNPDF3.0QED to this dataset. From Fig. 4.4 we observe that the PI contribution becomes smaller when using NNPDF3.1luxQED, though its effect is still comparable to the experimental uncertainties. Indeed, the PI contribution in the highest invariant mass bin ranges between 6% of the QCD cross-section in the central rapidity region and 1% for $|y_{ll}| = 2.5$. This comparison confirms that the PI contribution is important for the quantitative description of the Drell-Yan process above the $Z$ peak.

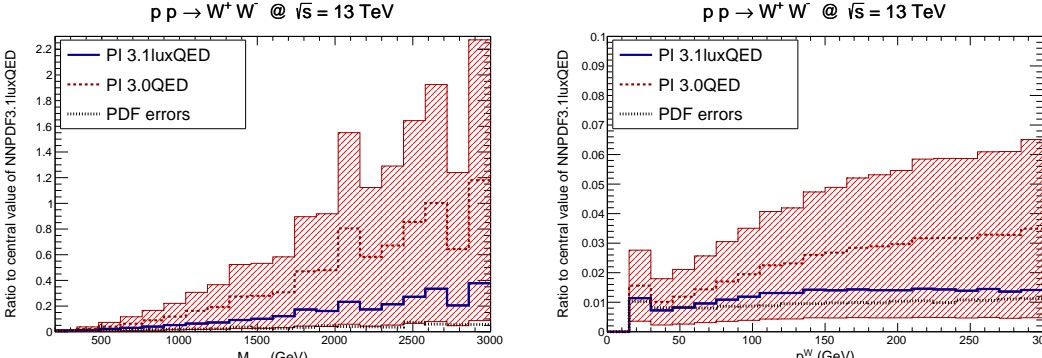

Figure 4.5: Same as Fig. 4.3 for the production of a $W^+W^-$ pair, specifically for invariant mass distribution $m_{WW}$ (left) and the transverse momentum of $W$ bosons $p_T^W$ (right plot).

## 4.2 Vector-boson pair production

The production of vector-boson pairs is of particular interest for the LHC physics program. Firstly, they probe the electroweak sector of the SM and provide bounds on possible anomalous couplings. Secondly, this final state appears in several BSM scenarios and therefore many searches involve the detection of pairs of weak vector bosons. When the invariant mass of the vector-boson pair $m_{VV}$ is large, the PI contributions, arising already at the Born level (see Fig. 4.1), are known to be significant [131]. Here we will examine the case of opposite-sign $W^+W^-$ production at the LHC 13 TeV.

To assess the size of the PI contribution to this process, in Fig. 4.5 we show the invariant mass distribution $m_{WW}$ and the transverse momentum distributions of the $W$ bosons $p_T^W$. The $W$ bosons are taken to be stable and required to be in the central rapidity region, $|\eta_W| \leq 2.5$. From the comparisons shown in Fig. 4.5 we find that for the $m_{WW}$ distribution the PI contribution is larger than the PDF uncertainties over the entire range considered. In particular, when using NNPDF3.1luxQED, the size of the PI contribution with respect to the total cross-section increases from 1% at $M_{WW} \simeq 300$ GeV up to 35% at $M_{WW} \simeq 3$ TeV. As expected, the trend is similar using NNPDF3.0QED but with much larger uncertainties. An upwards shift of the cross-section by a factor two or larger would be allowed within PDF uncertainties in this case.

The picture is rather different for the case of the transverse momentum distribution of the individual $W$ bosons $p_T^W$. Here we find that the PI contribution using NNPDF3.1luxQED are small, around the $\simeq 1\%$ level, over the entire range in $p_T^W$ considered. Additionally, the effect of using NNPDF3.0QED instead is not so dramatic, with an increase in the cross-section of a few percent at most. The differences between the two distributions arise from the fact that the PI contribution to $W$ boson pair production is kinematically enhanced only in the large $m_{WW}$ limit, irrespective of the value of $p_T^W$. The results of Fig. 4.5 suggest that current measurements of this process from ATLAS and CMS [132,133] might already be sensitive to the photon PDF.

## 4.3 Top-quark pair differential distributions

Next we turn to study the impact of the PI contributions on differential distributions in top-quark pair production (see also Refs. [5,51]). The APPLgrid tables generated for this process include only the $\gamma\gamma \to t\bar{t}$ channel. In Fig. 4.6 we show the ratio of the PI contribution over the QCD cross-section for the invariant mass distribution of top-quark pairs $m_{t\bar{t}}$ and the transverse momentum of the single top quarks $p_T^t$ at the 13 TeV LHC. Unlike in the case of high-mass Drell-Yan production, we find that the PI contribution to top-quark pair production is negligible even for the highest values of $m_{tt}$ and $p_T^t$ accessible at the LHC. Indeed, in the case of

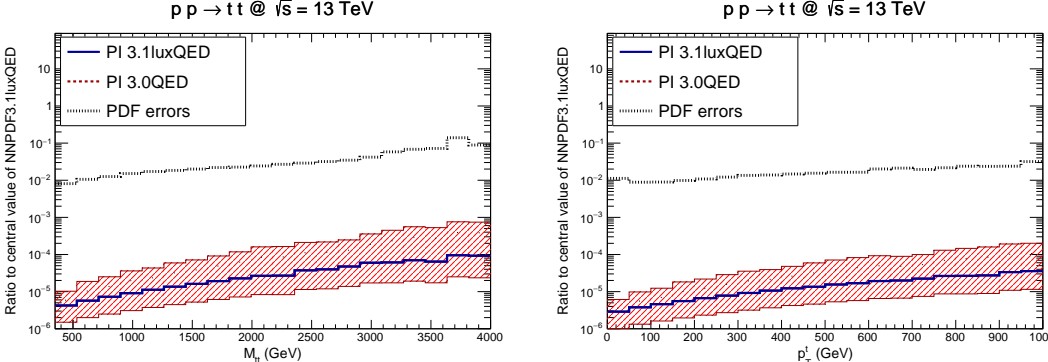

Figure 4.6: Same as Fig. 4.3 for the invariant mass distribution of top quark pairs $m_{t\bar{t}}$ (left) and the transverse momentum of top quarks $p_T^t$ (right plot) in top-quark pair production at 13 TeV.

NNPDF3.1luxQED the size of the PI contribution is at the permille level at most. Therefore, in theoretical calculations of top-quark pair production with electroweak corrections, the PI contribution can be safely neglected.

From the comparisons in Fig. 4.6 we also see that the PI correction is somewhat larger in NNPDF3.0QED but with larger associated uncertainties. Even in this case, at the highest invariant masses the upper edge of the 68% CL interval indicates that corrections due to the PI contribution are at most 0.1%. We have also verified that the PI contribution to $t\bar{t}$ production is phenomenologically negligible also for other distributions such as the rapidity distribution of top quarks and top-antitop pairs, $y_t$ and $y_{t\bar{t}}$, respectively.

## 4.4 Higgs production in association with a vector bosons

The last process that we consider in this section is Higgs production in association with a vector boson, see Fig. 4.1. PI corrections to this process are known to be significant. In the Higgs Cross Section Working Group prediction for the total cross-sections, where NNPDF3.0QED is used, the uncertainty due to the PI contribution is the dominant source of theory error [134]. To investigate this, we have generated and then combined exclusive samples for $pp \to hW^+$ and $pp \to hW^+j$ with and without the PI contribution. Both the Higgs boson and the $W^+$ boson are required to be in the central rapidity region, $|y_W| \leq 2.5$ and $|y_h| \leq 2.5$. No other kinematic cuts are applied.

In Fig. 4.7 we show the same comparison as in Fig. 4.2 for the Higgs transverse momentum $p_T^h$ and rapidity $y_h$ distributions. In the case of the $p_T^h$ distribution, we find that PI effects can be up to 5% when using NNPDF3.1luxQED, with the largest effects localised at intermediate values of $p_T^h \simeq 200$ GeV. We also note that the shift induced by the PI contribution is bigger than the PDF uncertainties. Concerning the $y_h$ rapidity distribution, the PI contribution can be $\simeq 6\%$ in the central rapidity region when using NNPDF3.1luxQED, while it becomes smaller as one moves to the forward region.

The comparisons of Fig. 4.7 illustrate that PI contributions are relevant also for Higgs boson physics, including the measurements of its couplings and branching fractions.

## 5 Summary

Parton distributions with QED effects and a photon PDF are an essential component in high-precision calculations of many LHC processes. Previous NNPDF QED sets adopted a data-

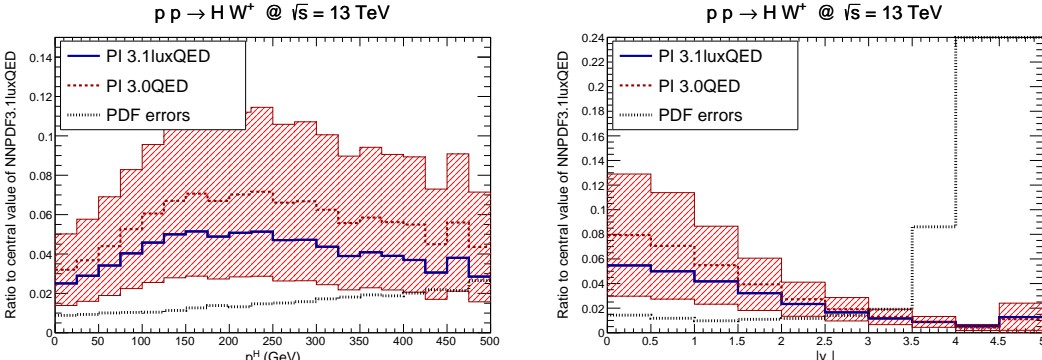

Figure 4.7: Same as Fig. 4.3 for Higgs production in association with a $W$ boson, for the Higgs transverse momentum $p_T^h$ distribution (left), and its rapidity $y_h$ distribution (right plot).

driven strategy to determine the photon PDF, independently parametrising $\gamma(x, Q_0)$ and then fitting it using constraints from Drell-Yan measurements at the LHC. While this strategy min-imised the theoretical bias due to model assumptions, the lack of a precise experimental handle to constrain the photon PDF led to large uncertainties.

With the development of the LUXqed framework, it is now possible to constrain the photon PDF in terms of the accurately known inclusive structure functions in lepton-hadron scatter-ing. In this work we have presented the NNPDF3.1luxQED set, where the photon content of the proton is determined by means of a global PDF analysis supplemented by the LUXqed the-oretical constraint. As a result, the uncertainty upon the photon PDF is considerably reduced as compared to our previous NNPDF3.0QED determination, down now to the level of a few percent. We find that photons carry up to 0.5% of the total momentum of the proton, and that the overall impact of the various types of QED effects included in NNPDF3.1luxQED induce small but non-negligible modifications in the quark and gluon PDFs.

We have then presented a first exploration of the implications of NNPDF3.1luxQED for photon-initiated processes at the LHC. We determine that the impact of PI contributions is consistent within uncertainties with respect to previous estimates based on NNPDF3.0QED except for the low-mass region $Q < M_Z$, and that they can be significant for many processes. For instance, we find corrections up to $\simeq 10\%$ for high-mass Drell-Yan and up to $\simeq 20\%$ for $W^+W^-$ production. In many cases, PI processes can be either comparable with or larger than PDF uncertainties. The uncertainty associated with these PI effects is in itself at the level of a few percent, so their overall effect is a shift of the cross-sections as compared to the QCD-only calculation.

The NNPDF3.1luxQED set represents a state-of-the-art determination of the PDFs of the proton including its photon component, accounting for all relevant theoretical and experimen-tal constraints. This set is therefore well suited for precision calculations of LHC processes. The NNPDF3.1luxQED sets are available via the LHAPDF6 interface [135]:

NNPDF31_nlo_as_0118_luxqed
NNPDF31_nnlo_as_0118_luxqed

while the FiatLux library, an open-source C++ implementation of the LUXqed formalism, can be obtained from:

https://github.com/scarrazza/fiatlux

together with the corresponding documentation.

**Acknowledgments**

We are grateful to our colleagues in the NNPDF Collaboration for many illuminating discussions on the photon PDF and fits with QED corrections, and specially to Stefano Forte for comments on this manuscript. We are grateful to Aneesh Manohar, Paolo Nason, Gavin Salam, and Giulia Zanderighi for discussions about the LUXqed formalism. J. R. is grateful to Lucian Harland-Lang for discussions about PDF fits with QED effects.

V. B., N. H., and J. R. are supported by an European Research Council Starting Grant "PDF4BSM". J. R. is also partially supported by the Netherlands Organization for Scientific Research (NWO). S. C. is supported by the HICCUP ERC Consolidator grant (614577) and by the European Research Council under the European Union's Horizon 2020 research and innovation Programme (grant agreement n° 740006).

# A    NNPDF3.1luxQED fit quality

In this Appendix we collect the results of the fit quality in the NNPDF3.1luxQED analysis and compare it to those from its QCD-only counterpart, NNPDF3.1. As customary in NNPDF analyses, these $\chi^2$ values are computed using the experimental definition of the covariance matrix, while the $t_0$ definition [136] was instead used during the fits in order to avoid the D'Agostini bias.

In Table A.1 we list the values of $\chi^2/N_{\text{dat}}$ from the NNPDF3.1 and NNPDF3.1luxQED NNLO fits for all the experiments included in the global analysis. We see that at the total dataset level, the fit quality is essentially identical in the two cases, yielding $\chi^2/N_{\text{dat}} = 1.148$. Also at the level of individual experiments there is good agreement between the two sets, with some small differences that are consistent with statistical fluctuations. Recall that these two fits are statistically independent, and therefore one expects fluctuations of the order $\Delta\chi^2 \sim \sqrt{N_{\text{dat}}}$ in the values of individual experiments.

From the comparison shown in Table A.1 we can conclude that while QED effects lead to small differences at the quark and gluon PDF level (see Fig. 3.5), the fit quality is still very similar to the QCD-only fit. This implies that these small QED effects can be reabsorbed into the PDFs, leading to the same quantitative description of the datasets included in the analysis. Note that the situation is likely to be rather different once measurements directly sensitive to the photon content of the proton, such as the ATLAS high-mass Drell-Yan at 8 TeV, are included into the fit.

Table A.1: The values of $\chi^2/N_{\mathrm{dat}}$ from the NNPDF3.1 and NNPDF3.1luxQED NNLO fits for all the experiments included in the global analysis. These $\chi^2/N_{\mathrm{dat}}$ values have been computed using the experimental definition.

| | $\chi^2/N_{\mathrm{dat}}$ | |
| | NNPDF3.1 | NNPDF3.1luxQED |
|---|---|---|
| NMC | 1.30 | 1.31 |
| SLAC | 0.75 | 0.71 |
| BCDMS | 1.21 | 1.21 |
| CHORUS | 1.11 | 1.11 |
| NuTeV dimuon | 0.82 | 0.75 |
| HERA I+II incl. | 1.16 | 1.16 |
| HERA $\sigma_c^{\mathrm{NC}}$ | 1.45 | 1.48 |
| HERA $F_2^b$ | 1.11 | 1.11 |
| DY E866 $\sigma_{\mathrm{DY}}^d/\sigma_{\mathrm{DY}}^p$ | 0.41 | 0.39 |
| DY E886 $\sigma^p$ | 1.43 | 1.43 |
| DY E605 $\sigma^p$ | 1.21 | 1.20 |
| CDF $Z$ rap | 1.48 | 1.48 |
| CDF Run II $k_t$ jets | 0.87 | 0.88 |
| D0 $Z$ rap | 0.60 | 0.60 |
| D0 $W \to e\nu$ asy | 2.70 | 2.68 |
| D0 $W \to \mu\nu$ asy | 1.56 | 1.57 |
| ATLAS total | 1.09 | 1.07 |
| ATLAS $W,Z$ 7 TeV 2010 | 0.96 | 0.96 |
| ATLAS HM DY 7 TeV | 1.54 | 1.57 |
| ATLAS low-mass DY 7 TeV | 0.90 | 0.88 |
| ATLAS $W,Z$ 7 TeV 2011 | 2.14 | 2.18 |
| ATLAS jets 2010 7 TeV | 0.94 | 0.91 |
| ATLAS jets 2.76 TeV | 1.03 | 1.02 |
| ATLAS jets 2011 7 TeV | 1.07 | 1.06 |
| ATLAS $Z$ $p_T$ 8 TeV $(p_T^{ll}, M_{ll})$ | 0.93 | 0.93 |
| ATLAS $Z$ $p_T$ 8 TeV $(p_T^{ll}, y_{ll})$ | 0.94 | 0.91 |
| ATLAS $\sigma_{tt}^{tot}$ | 0.86 | 0.89 |
| ATLAS $t\bar{t}$ rap | 1.45 | 1.22 |
| CMS total | 1.06 | 1.05 |
| CMS W asy 840 pb | 0.78 | 0.78 |
| CMS W asy 4.7 fb | 1.75 | 1.74 |
| CMS Drell-Yan 2D 2011 | 1.23 | 1.27 |
| CMS W rap 8 TeV | 1.01 | 1.00 |
| CMS jets 7 TeV 2011 | 0.84 | 0.81 |
| CMS jets 2.76 TeV | 1.03 | 1.02 |
| CMS $Z$ $p_T$ 8 TeV $(p_T^{ll}, y_{ll})$ | 1.32 | 1.32 |
| CMS $\sigma_{tt}^{tot}$ | 0.20 | 0.16 |
| CMS $t\bar{t}$ rap | 0.94 | 0.99 |
| LHCb total | 1.47 | 1.47 |
| LHCb $Z$ 940 pb | 1.49 | 1.48 |
| LHCb $Z \to ee$ 2 fb | 1.14 | 1.11 |
| LHCb $W,Z \to \mu$ 7 TeV | 1.76 | 1.79 |
| LHCb $W,Z \to \mu$ 8 TeV | 1.37 | 1.36 |
| **Total** | **1.148** | **1.148** |

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
