# Peer review of "Illuminating the photon content of the proton within a global PDF analysis"

_SciPost Physics, doi:SciPost Phys. 5, 008 (2018)_

## Round 1 · Referee Report · Anonymous · 2018-1-25

Strengths

1) This paper incorporates the precise LUXqed description of the photon within the NNPDF global PDF framework. As such, it represents the state-of-the-art description of the photon content within the proton.

2) The correct inclusion of photon-initiated processes is an important issue for precision LHC phenomenology, about which there has been much confusion in the past. These results will now represent the standard for use with the NNPDF set, allowing such processes to be dealt with precisely and consistently in the future, as they must be.

3) In addition to describing the implementation of the LUXqed photon within NNPDF, a range of phenomenological results are presented and discussed. This is a very useful exercise and will no doubt guide future phenomenology.

Weaknesses

The weaknesses of the paper are addressed in my point by point list of corrections below.

Report

In general this represents an important contribution to the field. I find the paper to be clear and comprehensive, and the efforts made to consider a range of phenomenological applications particularly useful. However, there are number of issues which I believe need to be addressed before I can recommend it for publication.

Requested changes

1) Page 3, first full paragraph. The discussion of the approach to calculating the photon PDF based on a theoretically motivated ansatz gives undue prominence to CT14QED. While it is true that the final public release of the CT14QED analysis includes the elastic component of the photon PDF, the original study did not consider this. The inclusion of the elastic component came almost a year after the original release, and came subsequently to the discussion of [33,34]. In addition, Refs [30-34] all include a model of the inelastic component, and so are qualitatively no different in approach from the CT14QED set. Given that the idea of this introductory paragraph is to describe the model dependent approach, CT14QED and Refs [30-34] should be dealt with on a more equal footing, ideally giving some indication of how these ideas have developed chronologically.

2) Page 3, last sentence of second paragraph. English-wise this needs a little rewording: "Although this dataset is particularly...". It would also perhaps be fairer to [39] to say that some reduction in uncertainty is achieved relative to the baseline.

3) Page 3. The point should be made somewhere here that the elastic component is by far the dominant contribution to the input photon distribution, in particular at higher x, and thus the uncertainties are already greatly reduced by including this. This is even briefly discussed later on at the end of section 2.4, but is easily missed there.

4) Page 3, third full paragraph. Here or somewhere else, the earlier works (Anlauf et al. Comput.Phys.Commun. 70 (1992) 97-119, Mukherjee and Pisano Eur.Phys.J. C30 (2003) 477-486, Blumlein et al J.Phys. G19 (1993) 1695-1703) should be referenced to. These independently calculated expressions using a similar approach to LUXqed, i.e. relating the inelastic photon to the proton structure functions. These resulted in expressions for the photon that were very close to the LUXqed result, with the exception of the limits on the Q^2 integral, which were not correct, and the missing mass correction in the Blumlein case. Clearly LUXqed represents the state of the art in this respect, but a reference and brief description would be fair.

5) Page 3, second to last paragraph. The statement that the new photon is 'fully consistent' is not supported by the current results, even when phrased in terms of the impact of the photon PDF. From Fig. 4.3 (also 3.3) we can see that the 3.1lux photon-initiated contribution is important relative to other PDF uncertainties, but also inconsistent with the earlier 3.0 prediction. I discuss this more below, but this cannot be the right thing to say here if this result stands.

6) Section 2.3, third paragraph. It would be useful to show a plot of the impact of the higher order corrections on the photon-photon luminosity.

7) Section 2.4, below (2.2). `A fraction of its uncertainty' seems a little vague. What fraction is taken?

8) Page 8. Ref [116] should be supplemented with 1601.03413, which came before this study (both are referenced in the LUXqed paper).

9) Start of section 3.1. Unless I have missed it, the difference between LUXqed 16 and 17 does not seem to be described anywhere in the paper. It would make sense to do this at some point.

10) Figure 3.1. For the purposes of comparing the cases with only the high Q^2 uncertainties vs. the full case, things are perhaps not presented in the best way. I think it would be helpful to have both cases on the same plot in some way, so that the differences can be seen more directly, but this is not essential.

11) Page 9, first paragraph. Perhaps it is worth clarifying a little where the dependence on the perturbative order is expected to occur? Surely, at least to first approximation, the only dependence on this comes from the high Q^2 component, as the other LUX components are independent of order?

12) Fig 3.3 (and 3.4 by implication) and Page 9, second paragraph. The fact that 3.0 photon undershoots the 3.1luxQED photon at low x is surely surprising. In particular, at high scales and low x (i.e. Fig. 3.3. right) the photon is entirely driven by perturbative DGLAP, i.e. in terms of the other partons. Given the compatibility of the 3.0 and 3.1 quark/gluon PDFs, I do not understand how the photon PDFs can look so different in this region. Might it be that the 3.0 photon is calculated using the 2.3 evolution procedure (subsequently corrected)? In any case, given the size of the difference relative to the PDF uncertainties of NNPDF3.0QED the reason for this apparent discrepancy has to be discussed.

This all feeds through to Fig. 3.7, where again no explanation for the tension at low mass is discussed. Then again in Section 4, differences in various predictions at lower mass are seen for the same reason, but not discussed.

13) Fig 3.3. Did the authors intend to take an absolute plot on the left and ratio plot on the right? Given ratios are considered everywhere else it would be more consistent to take that on the left, but clearly this is a minor point.

14) Section 4. How is the scale of \alpha treated for the coupling of the initial state photons in the matrix element? Historically many people have wrongly used \alpha(0), but as discussed in 1605.04935 and 1705.00598 this is not appropriate for the case of initial state photons with corresponding photon PDFs, even though these are treated as on-shell in the matrix elements; instead \alpha(\mu_F) should be taken. The scale choice should be mentioned, and if the on-shell coupling is used, corrected.

15) Fig. 4.3. Perhaps it is worth emphasising the difference in scale on the y axis relative to the other plots at some point, just for clarity.

16) Page 16, first full paragraph. Typos- should be 300 GeV and M_ll.

17) Page 18, last full paragraph. Again the statement about consistency with respect to 3.0 should be rephrased in light of the discussion above.

  • validity: good
  • significance: top
  • originality: good
  • clarity: high
  • formatting: excellent
  • grammar: excellent

Author:  Stefano Carrazza  on 2018-02-13  [id 214]

(in reply to Report 1 on 2018-01-25)

We thank the referee for the close reading of our manuscript and for the detailed report which will help us greatly in clarifying various issues.
We have addressed the various comments raised by the referee in the PDF file in attachment and an updated version of the manuscript is ready for resubmission.

Attachment:

referee_reply.pdf

---

## Round 1 · Referee Report · Maxime Gouzevitch · 2018-3-27

Strengths

1) The topic is highly relevant for precision LHC physics.

2) This is the first attempt to use the groundbreaking formalism of LUXqed to extract the photon PDF from a fully coherent PDF fit within a well known NNPDF framework.

3) This paper opens a way to a combined LUXqed + Photon-PDF-sensitives LHC data fit that would be the state of the art of the topic.

Weaknesses

1) It is not clear why there was no attempts to use LUXqed formalism with NNPDF3.0 dataset that was used to extract NNPDF3.0QED. This dataset was not stripped from photon PDF sensitive data and already included Run I DY samples. Please explain.

2) There is a disagreement between NNPDF3.1LUXqed and NNPDF3.0 at low x that has few sigma significance. Some explanations shall be provided. Using NNPDF3.0QED setup with LUXQED could have helped to understand it. Is there some underestimated systematic in NNPDF3.0QED fit?

Report

The paper if an important milestone on the way to understand the photon content of the proton. This minor contribution (< 1%) was neglected till now. The experimental reality of the LHC made this contribution shine and required a quantification. From one hand we know that the LHC appeared to be a wonderful gluon-gluon collider producing large number of ttbar and Higgs events. While the gluon content of the proton completely outnumber the one of the photon, the average momentum of the gluons is smaller because of the gluons self-coupling ability and large color charge. From the other hand high precision measurements in Drell-Yann production are sensitive to % level effects and the photons contribution is experimentally observable in off-shell DY production -- just outside of the Z peak or at very high DY mass.

The first attempts to constraint the photon PDF were based on classical PDF fitting approach where a new photon PDF was added constrained by the sum rules and the LHC data. The impact of this approach appeared to be limited (50-100% precision on the photon PDF). An alternative and elegant approach, LUXqed formalism, related the F2 and FL functions accurately measured in DIS to the photon PDF. The fundamental idea is that in DIS a photon is exchanged between the lepton and the proton and this process is therefore related to the collinear content in photons of the proton. This latter was first tried using the PDF4LHC reweighting procedure to assess the photons content.

In this paper we have the first fully coherent PDF extraction of the photon PDF using LUXqed formalism and a global dataset (fixed target, DIS and hardon-hadron collider experiments). It is important to notice that this is not yet the optimal solution. Indeed the dataset used for NNPDF3.1 was not designed to measure the photon PDF. The data samples was chosen in such a way to reduce the sensitivity to the EW effects not included into the fits. The authors discuss the compatibility between LUXqed and previous "direct" extractions of the photon PDF and project in future to combine both direct and indirect constraints in the next generation of NNPDF.

Requested changes

Section 2.2:

1) This is a classical theoretical papers bias that we try to fight in experimental publications. Any variable used have to be defined: z, x, mu etc... Sometimes in experiments we use different notations and it requires time to guess the meaning of each variable.
Same point for section 2.5 page 8. RL/T is not defined.

Section 2.3:

2) "Charm PDF is fitted on equal footing as light ..." --> It means that the charm PDF is parametrized at Q0? But this is not the case for b-quark? Please be more clear.

Section 3.1:

3) I would suggest to spend some lines to better describe the assumptions behind LUXqed16/17 and NNPDF3.0QED you have a long discussion of comparing them and not everybody have time to read carefully the papers you refer to (some of the papers like NNPDF3.0 one is 150 pages long ;) ).

4) Page 9: I would like to see a discussion about the disagreement between NNPDF3.1LUXqed and NNPDF3.0 at low x.

Section 4 page 14:

5) The argument that all calculations are using LO MC with NNLO PDF is not very clear to me. I understand that using NLO or NNLO PDF is more or less equivalent since QCD effects are small. But usually LO PDFs have much larger gluon content. Please explain.

6) Figure 4.5: You may like to extend the range up to pTWW = MWW/2 ~ 1.5 TeV. Then you would probably see around pT ~ 1.5 TeV the same kind of effect you see at 3 TeV in MWW.

7) Section 4.3: please state if you include only diagram 3 from figure 4.1 or also gamma gamma -> ttbar

8) I would like to see a discussion why

(gamma gluon --> ttbar)/(gg->ttbar) << 1

while

(gamma gamma -> ll)/ DY < 1

Is it related to the fact that for DY QED contribution is significant only for off-shell Z production where diagram 1 figure 4.1 is significant since it is t-channel ? And for ttbar the main gluon-gluon diagram is already t-channel?

9) page 3: "Overcoming the limitations both two strategies" --> Overcoming the limitations of both strategies.

10) Figure 4.2: Add please the NNPDF3.0QED uncertainty band to the legend

11) Figure 4.3: It is not exactly the same. You take the right figure from 4.2 and repeat it twice for high and low mass. Please change a bit the legend to make it less disturbing for the reader.

  • validity: top
  • significance: high
  • originality: top
  • clarity: high
  • formatting: good
  • grammar: good

Anonymous on 2018-04-16  [id 244]

(in reply to Report 2 by Maxime Gouzevitch on 2018-03-27)
Category:
answer to question

We thank the referee for the close reading of our manuscript and for the detailed report which will help us greatly in clarifying various issues.
We have addressed the various comments raised by the referee in the PDF file in attachment and an updated version of the manuscript is ready for resubmission.

Attachment:

referee2_reply.pdf

---

## Round 2 · Referee Report · Anonymous · 2018-5-11

Strengths
As stated in my original report.
Weaknesses
There remains an issue with the comparison between the 3.0 and new Lux set which as far as I can see has not been resolved, while consistency between these two is still claimed.
Report
Firstly, I would like to apologise for the long delay in my responding. I did not receive a notification from the journal of your reply, and so missed this for some time. I would have replied much earlier otherwise.
I would like to thank the authors for their careful response to my comments. However, I feel I have to insist upon this issue of the NNPDF3.0 vs. 3.1LUXqed comparison before I can personally recommend publication.
Two reasons are given for this difference. The first reason, namely the impact of higher-order corrections in the evolution, is clearly quite small and certainly (as the authors acknowledge) far too small to explain the size of the difference. This leaves the second stated reason, namely the fact that the input is due to the 2.3 fit, which used a different form of evolution to the current unified one. In other words, the claim is that if the 2.3 fit were redone, but using the unified (3.0) evolution, the NNPDF3.0 and 3.1LUXqed would (at least largely) converge. I have two objections to this:
1) Even if we accept the logic as described, this only remains a possible reason, as no such 2.3 fit, but now with unified evolution, has been performed. Perhaps if this were done the discrepancy would remain; from the information we have, we (at best) simply do not know. I am not suggesting this is done (that would clearly be beyond the scope of the paper), but nonetheless there is still no reliable basis to claim, as is still currently done in the introduction and conclusion, that the updated PDF set is fully consistent within uncertainties with NNPDF3.0QED.
2) My stronger objection is that as far as I can tell the logic described by the authors would actually lead to an increase in the tension between NNPDF3.0 vs. 3.1LUXqed. In particular, as the unified evolution is faster at low x in comparison to that used in the 2.3 set, if this fit were to be repeated using the unified evolution, we would surely require a *smaller* and not a larger, input photon at low x to fit the same data, i.e. to compensate the faster evolution. In other words, the tension would if anything increase.
Given this, it appears to me that for whatever reason the previous NNPDF agnostic QED fits do not give results at low x which are consistent with 3.1LUXqed, for a reason which cannot be due to this differing evolution (or at the very least has not been shown to be), but rather something relating to the earlier fit itself. The reason for this remains unclear, but consistency definitely cannot be claimed.
I look forward to receiving further comments on this, if I am mistaken; though even if I am in terms of point (2), point (1) remains. So, the comments claiming consistency between 3.0 and 3.1LUXqed should be rephrased - any consistency cannot currently be claimed - and the discussion around Fig. 3.3 updated. Ideally reasons for this apparently genuine disagreement would be considered, but I appreciate at this stage this may not realistic.
Requested changes
As described above.
Anonymous on 2018-05-24 [id 257]
(in reply to Report 3 on 2018-05-11)First of all, we fully agree with the referee that we cannot really claim consistency between NNPDF3.0QED and NNPDF3.1LUXqed in the small-x region below roughly x=0.01. Irrespective of the reason, this is just an empirical fact that the two sets do not agree within the quoted uncertainties in this region. We have modified the text to reflect better this fact, which perhaps was not that clearly stated in the previous revision of the paper. We have also added a remark that for the region more relevant for LHC phenomenology, namely x > 0.01 (which roughly corresponds to Q > MZ), both sets agree indeed within uncertainties.
Concerning point 1), we have toned down the statements concerning the possible explanation for the discrepancy between NNPDF3.0 and NNPDF3.1luxQED. As the referee rightly points out, without redoing the same fit with the correct evolution settings (and comparing exactly like with like) it is not really possible to categorically conclude that this is the dominant reason for the differences. So now we state that this is a possible reason (certainly it does contribute to the differences) but we don't know for sure that this is the main one, since one could conceive other options, due to the radically different methodologies adopted in the two sets. As the referee acknowledges, a final clarification of this point would be beyond the scope of this paper.
Concerning point 2), it seems to us that the situation is a bit more complex than that. Indeed, the NNPDF2.3QED photon PDF was constrained mostly by Drell-Yan data at the LHC (since the constraints only from DIS structure functions were extremely loose). Therefore the right comparison would be at the level of PDF luminosities, rather than at the level of PDFs themselves. Note for example that in Fig. 3 of arXiv:1606.07130 the effects of the 3.0QED evolution actually lead to a small suppression of the photon PDF at large-x rather than an enhancement. This would be relevant for say the LHCb low-mass Drell-Yan data used in the NNPDF2.3QED fit, where one parton is at low-x and the other at small-x. So one should look at the kinematics of all hadronic datasets used in NNPDF2.3QED point by point to see how these differences in the evolution feed into the resulting photon, and this is rather complicated without doing the full fit. All in all, the only reason to conclusively set this point would be to redo NNPDF2.3QED with the same evolution as NNPDF3.0QED.
In any case, as mentioned above, we agree with the referee that one cannot exclude other sources of difference between NNPDF2.3/3.0QED and NNPDF3.1luxQED, perhaps of methodological origin. We have made sure that this fact is properly reflected in the text so that there are no ambiguities.
We hope that with the modifications that we have carried out in the new revision of the paper, the referee will consider it suitable for publication in SciPost.
Anonymous on 2018-05-28 [id 261]
(in reply to Anonymous Comment on 2018-05-24 [id 257])Thank you for this update, I think this describes the situation more clearly. As you say, empirically one cannot claim 'consistency' in the sense that the PDF sets do indeed not agree for all x regions.
The remaining question is whether the sets are 'consistent' in the sense that, if the same theoretical framework for the evolution were used throughout, the results would be consistent. The point about the suppression at high x for the 3.0 (vs. 2.3) evolution is certainly a good one, although I personally still find it very hard to see how this sort of effect, which is clearly relatively mild, could lead to such a sizeable suppression in the 3.0QED photon at low x. Somehow it seems that there is something else methodological in the original 2.3QED case (after all a reweighting to a rather unconstraining dataset as we know) at play here. I may be wrong though.
In any case as we are all agreed this is beyond the scope of the current paper. I am completely happy with the updated version and offer no further comments or suggestions.

---

## Round 2 · Author Response

You are currently on this page

---

## Round 3 · Author Response

Updated version incorporating last changes requested by referee.

---

## Round 3 · List of Changes

As listed in the last referee reply.

---

## Editorial Decision

published